# Learning Accurate Decision Trees with Bandit Feedback via Quantized Gradient Descent

**Ajaykrishna Karthikeyan**\*† *ak76@illinois.edu*
*Department of Computer Science*
*University of Illinois, Urbana-Champaign*

**Naman Jain**\*† *naman_jain@berkeley.edu*
*Department of Electrical Engineering and Computer Science*
*University of California, Berkeley*

**Nagarajan Natarajan** *nagarajn@microsoft.com*
*Microsoft Research, India*

**Prateek Jain**† *prajain@google.com*
*Google Research, India*

**Reviewed on OpenReview:** *https://openreview.net/forum?id=uOn1chY0b6*

## Abstract

Decision trees provide a rich family of highly non-linear but efficient models, due to which they continue to be the go-to family of predictive models by practitioners across domains. But learning trees is challenging due to their discrete decision boundaries. The state-of-the-art (SOTA) techniques resort to (a) learning *soft* trees thereby losing logarithmic inference time; or (b) using methods tailored to specific supervised learning settings, requiring access to labeled examples and loss function. In this work, by leveraging techniques like overparameterization and straight-through estimators, we propose a unified method that enables accurate end-to-end gradient based tree training and can be deployed in a variety of settings like offline supervised learning and online learning with bandit feedback. Using extensive validation on standard benchmarks, we demonstrate that our method provides best of both worlds, i.e., it is competitive to, and in some cases more accurate than methods designed *specifically* for the supervised settings; and in bandit settings, where most existing tree learning techniques are not applicable, our models are still accurate and significantly outperform the applicable SOTA methods.

## 1 Introduction

Decision trees are an important and rich class of non-linear ML models, often deployed in practical machine learning systems for their efficiency and low inference cost. They can capture fairly complex decision boundaries and are state-of-the-art (SOTA) in several application domains. Optimally learning trees is a challenging discrete, non-differentiable problem that has been studied for several decades and is still receiving active interest from the community (Gouk et al., 2019; Hazimeh et al., 2020; Zantedeschi et al., 2021).

Most of the existing literature studies tree learning in the context of traditional supervised settings like classification (Carreira-Perpinán and Tavallali, 2018) and regression (Zharmagambetov and Carreira-Perpinan, 2020), where the methods require access to explicit labeled examples and full access to a loss function. However, in several scenarios we might not have access to labeled examples, and the only supervision available might be in the form of evaluation of the loss/reward function at a given point. One such scenario is the

---

\* Equal Contribution
† Work done while author was at Microsoft Research, India

"bandit feedback" setting, that often arises in real-world ML systems (e.g., recommender systems with click-feedback), where the feedback for the deployed model is in the form of loss/reward. The bandit setting has been long-studied both in the theoretical/optimization communities as well as in the ML community (Dani et al., 2008; Dudík et al., 2011; Agarwal et al., 2014; Shamir, 2017). The Vowpal Wabbit (VW) library is extensively used by practitioners for solving learning problems in this setting.

So, in this work, we address the question: *can we design a general method for training hard[1] decision trees, that works well in the standard supervised learning setting, as well as in settings with limited supervision?* In particular, we focus on online bandit feedback in the presence of context. Here, at each round the learner observes a context/example, picks an action/prediction either from a discrete space (classification) or continuous space (regression) for which it then receives a loss, which must be minimized. Since the structure of this loss function is unknown to the learner, it is referred to as black-box loss, in contrast to the loss observed in the supervised setting where the entire structure is known to the learner. Notice that this loss function can be evaluated only *once* for an example. Hereafter, we will refer to this setting in the paper simply as the *bandit setting* or the *bandit feedback setting*.

Recently, several neural techniques have been proposed to improve accuracy of tree methods by training with differentiable models. However, either their (Yang et al., 2018; Lee and Jaakkola, 2020) accuracies do not match the SOTA tree learning methods on baseline supervised learning tasks or they end up with "soft" decision trees (Irsoy et al., 2012; Frosst and Hinton, 2017; Biau et al., 2019; Tanno et al., 2019; Popov et al., 2020) which is not desirable for practical settings where inference cost may be critical. Using annealing techniques to harden the trees heavily compromises on the accuracy (as seen in Section 5). On the other hand, there is a clear trade-off between *softness* (and in turn inference cost) and performance in SOTA tree methods (Hazimeh et al., 2020) which try to minimize the number of leaves reached per example (Section 5).

SOTA (*hard*) tree learning methods (Carreira-Perpinán and Tavallali, 2018; Hehn et al., 2019; Zharmagambetov and Carreira-Perpinan, 2020), based on alternating optimization techniques, yield impressive performance in the batch/fully supervised setting (as seen in Section 5), but their applicability drastically diminishes in more general bandit settings where the goal is to achieve small *regret* with few queries to the loss function. It is unclear how to extend such methods to bandit settings, as these methods optimize over different parameters alternatively, thus requiring access to multiple loss function evaluations per point.

We propose a simple, end-to-end gradient-based method "Dense Gradient Trees" (DGT) [2] that gives best of both worlds — (1) logarithmic inference cost, (2) significantly more accurate and sample-efficient than general methods for online settings, (3) competitive to SOTA tree learning methods designed specifically for supervised tasks, and (4) easily extensible to ensembles that outperform widely-used choices by practitioners. DGT appeals to the basic definition of the tree function that involves a product over *hard* decisions at each node along root-to-leaf paths, and $\arg\max$ over paths. We make three key observations regarding the tree function and re-formulate the learning problem by leveraging successful techniques from deep networks training literature: 1) the path computation AND function can be expressed in "sum of decisions" form rather than the multiplicative form which has ill-conditioned gradients, 2) we can overparameterize the tree model via learning a *linear deep embedding* of the data points without sacrificing efficiency or the tree structure, and 3) hard decisions at the nodes can be enforced using quantization-aware gradient updates.

The reformulated tree learning problem admits efficient computation of *dense* gradients with respect to model parameters leveraging quantization-aware (noisy) gradient descent and gradient estimation via perturbation. We provide algorithms for standard supervised learning (DGT) and bandit (DGT-Bandit) settings, and extensively evaluate them on multiple classification and regression benchmarks, in supervised as well as in bandit settings. On supervised tasks, DGT achieves SOTA accuracy on most of the datasets, comparing favorably to methods designed specifically for the tasks. In the bandit setting, DGT-Bandit achieves up to 30% less regret than the SOTA method for classification and up to 50% less regret than the applicable baseline for regression.

---

[1]Informally, a *hard* tree is one where during inference, an input example decisively moves down to a single child at an internal node, thus landing on a single leaf finally. In contrast, an input example in a *soft* tree is probabilistically distributed over children of an internal node, and hence over leaves of the tree as well.

[2]Source code is available at `https://github.com/microsoft/dgt`.

Our key contributions are: **1)** A unified and differentiable solution for learning decision trees accurately for practical and standard evaluation settings, **2)** competitive (or superior) performance compared to SOTA hard decision tree learning methods on a variety of datasets, across different problem settings, and **3)** enabling sample-efficient tree learning in the online, bandit feedback setting without exact information on the loss function or labels.

## 1.1 Related Work

Several lines of research focus on different aspects of the tree learning problem, including shallow and sparse trees (Kumar et al., 2017), efficient inference (Jose et al., 2013), and incremental tree learning in streaming scenarios (Domingos and Hulten, 2000; Jin and Agrawal, 2003; Manapragada et al., 2018; Das et al., 2019) (where labels are *given* but arrive online). The most relevant ones are highlighted below.

**Non-greedy techniques:** "Tree Alternating Optimization" (TAO) is a recent SOTA non-greedy technique for learning classification (Carreira-Perpinán and Tavallali, 2018) and regression (Zharmagambetov and Carreira-Perpinan, 2020) trees. It updates the nodes at a given height (in parallel) using the examples routed to them in the forward pass while keeping the nodes in other heights fixed. The gradients during backpropagation are "sparse" in that each example contributes to $O(h)$ updates. This poses a challenge from a sample complexity perspective in the online setting. In contrast, DGT performs "dense" gradient updates, i.e., with one example's one (quantized) forward pass, $O(2^h)$ model parameters can be updated in our backpropagation. Interestingly, we find DGT achieves competitive results to TAO even in the batch setting (see Section 5). Norouzi et al. (2015) optimizes a (loose) relaxation of the empirical loss; it requires $O(h)$ loss evaluations *per example* (as against just 1 in DGT) to perform gradient update, so it does not apply to bandit settings. Bertsimas and Dunn (2017); Bertsimas et al. (2019) formulate tree learning as a mixed integer linear program (ILP) but their applicability is limited in terms of problem settings and scale. LCN (Lee and Jaakkola, 2020) uses $h$-layer ReLU networks to learn oblique trees of height $h$. But, for a given height, their model class admits only a subset of all possible trees which seems to deteriorate its performance (see Section 5).

**Soft trees/Neural techniques:** Several, mostly neural, techniques (Jordan and Jacobs, 1994; Irsoy et al., 2012; Kontschieder et al., 2015; Balestriero, 2017; Frosst and Hinton, 2017; Biau et al., 2019; Tanno et al., 2019; Popov et al., 2020) try to increase the accuracy of decision trees by resorting to soft decisions and/or complex internal nodes and end up producing models that do not have oblique tree structure (Definition 1). The recently proposed (Zantedeschi et al., 2021) embeds trees as layers of deep neural networks, and relaxes mixed ILP formulation to allow optimizing the parameters through a novel "argmin differentiation" method. In Jordan and Jacobs (1994); Irsoy et al. (2012); Frosst and Hinton (2017), all root-to-leaf path probabilities have to be computed for a given example, and the output is a weighted sum of decisions of all paths. Two recent tree learning methods are notable exceptions: 1) Hehn et al. (2019) uses an E-M algorithm with sigmoid activation and annealing to learn (hard) trees, and 2) TEL (Hazimeh et al., 2020) is a novel smooth-step activation function-based method that yields a *small* number of reachable leaves at the end of training, often much smaller than $O(2^h)$ for probabilistic trees. However, the inference cost of the method is relatively much larger than ours, to achieve competitive accuracy, as we show in Section 5.3.

**Bandit feedback:** Contextual bandit algorithms for this setting allow general reward/loss functions but are mostly applicable to discrete output domains like classification (Agarwal et al. (2014)); e.g. Féraud et al. (2016) learns an axis-aligned decision tree/forest on discretized input features, and Elmachtoub et al. (2017) learns a separate decision tree for every action via bootstrapping. In contrast, DGT uses noisy gradient estimation techniques (Flaxman et al., 2005; Agarwal et al., 2010) (together with our tree learning reformulation) to provide an accurate solution for both bandit classification and regression settings.

## 2 Problem Setup

Denote data points by $\mathbf{x} \in \mathbb{R}^d$. In this work, we focus on learning oblique binary decision trees.

**Definition 1** (Oblique binary decision tree). *An oblique binary tree of height $h$ represents a piece-wise constant function $f(\mathbf{x}; \mathbf{W}, \mathbf{\Theta}) : \mathbb{R}^d \rightarrow \mathbb{R}^K$ parameterized by weights $\mathbf{w}_{ij} \in \mathbb{R}^d, b_{ij} \in \mathbb{R}$ at $(i, j)$-th node ($j$-th node at depth $i$) computing decisions of the form $\langle \mathbf{w}_{ij}, \mathbf{x} \rangle + b_{ij} > 0$, that decide whether $\mathbf{x}$ must traverse the*

*left or right child next. For classification, we associate parameters $\boldsymbol{\theta}_j \in \mathbb{R}^K$ at $j$-th leaf node, encoding scores or probabilities for each of the $K$ classes. For regression, $\theta_j \in \mathbb{R}$ ($K = 1$) is the prediction given by the $j$-th leaf node.*

We devise algorithms for learning tree model parameters $(\mathbf{W}, \boldsymbol{\Theta})$ in two different settings.

**(I) Contextual Bandit setting.** In many practical settings, e.g. when ML models are deployed in online systems, the learner gets to observe the features $\mathbf{x}_i$ but may not have access to label information; the learner only observes a loss (or reward) value for its prediction on $\mathbf{x}_i$. We have zeroth-order access to $\ell : f(\mathbf{x}_i; \mathbf{W}, \boldsymbol{\Theta}) \to \mathbb{R}_+$, i.e. $\ell$ is an *unknown* loss function that can only be queried at a given point $(\mathbf{W}_i, \boldsymbol{\Theta}_i)$. The goal then is to minimize the *regret*:

$$\frac{1}{n} \sum_{i=1}^{n} \ell\big(f(\mathbf{x}_i; \mathbf{W}_i, \boldsymbol{\Theta}_i)\big) - \min_{\mathbf{W}, \boldsymbol{\Theta}} \frac{1}{n} \sum_{i=1}^{n} \ell\big(f(\mathbf{x}_i; \mathbf{W}, \boldsymbol{\Theta})\big).$$

**(II) Supervised learning.** This is the standard setting in which, given labeled training data $(\mathbf{x}_i, y_i)_{i=1}^{n}$, where $y_i$ is the label of $\mathbf{x}_i$, and a *known* loss $\ell : \big(f(\mathbf{x}; \mathbf{W}, \boldsymbol{\Theta}), y\big) \to \mathbb{R}_+$, the goal is to learn a decision tree that minimizes empirical loss on training data:

$$\min_{\mathbf{W}, \boldsymbol{\Theta}} \quad \frac{1}{n} \sum_{i=1}^{n} \ell\big(f(\mathbf{x}_i; \mathbf{W}, \boldsymbol{\Theta}), y_i\big) + \lambda \Phi_{\text{reg}}(\mathbf{W}, \boldsymbol{\Theta}), \tag{1}$$

where $\Phi_{\text{reg}}(.)$ is a suitable regularizer for model parameters, and $\lambda > 0$ is the regularization parameter. We consider both regression (with squared loss) and classification (with multiclass 0-1 loss).

Our method DGT (Section 4) provides an end-to-end gradient of tree prediction with respect to the tree parameters, so we can combine the tree with any other network (i.e., allow $f(g(\mathbf{x}); \mathbf{W}, \boldsymbol{\Theta})$, where $g$ and $\boldsymbol{\Theta}$ can be neural networks) or loss function, and will still be able to define the backpropagation for training the tree. So, in addition to the above loss functions and supervised learning setting, DGT readily applies to a more general set of problems like multi-instance learning (Dietterich et al., 1997), semi-supervised learning (Van Engelen and Hoos, 2020), and unsupervised learning (Chapter 14, Friedman et al. (2001)). We also consider a forest version of DGT (called DGT-Forest) in Section 5.

**Notation**: Lowercase bold letters $\mathbf{x}, \mathbf{w}$, etc. denote column vectors. Uppercase bold letters (e.g. $\mathbf{W}, \boldsymbol{\Theta}$) denote model parameters. $\mathbf{x}(i), \mathbf{w}(j)$ denote the $i$-th and the $j$-th coordinate of $\mathbf{x}$ and $\mathbf{w}$, respectively.

## 3  Our Tree Learning Formulation

Learning decision trees is known to be NP-hard (Sieling, 2003) due to the highly discrete and non-differentiable optimization problem. SOTA methods typically learn the tree greedily (Breiman et al., 1984; Quinlan, 2014) or using alternating optimization at multiple depths of the tree (Carreira-Perpinán and Tavallali, 2018; Zharmagambetov and Carreira-Perpinan, 2020). These methods typically require full access to the loss function (see Related Work for details). We remedy this concern by introducing a simple technique that allows end-to-end computation of the gradient of all tree parameters. To this end, we first introduce a novel flexible re-formulation of the standard tree learning problem; in Section 4, we utilize the re-formulation to present efficient and accurate learning algorithms in both bandit and supervised (batch) settings.

Consider a tree of height $h$. Let $I(i, l)$ denote the index of the $l$-th leaf's predecessor in the $i$-th level of the tree; e.g., for all $i$, $I(i, l) = 0$ if $l$ is the left-most leaf node. Also, define $S(i, l)$ as:

$$S(i, l) = \begin{cases} -1, & \text{if } l\text{-th leaf} \in \text{left subtree of node } I(i, l), \\ +1, & \text{otherwise.} \end{cases}$$

Recall that $\mathbf{w}_{ij} \in \mathbb{R}^d$ and $b_{ij} \in \mathbb{R}$ are the weights and bias of the node $j$ at depth $i < h$, and $\boldsymbol{\theta}_l$ are the parameters at leaf $l$. Now, the decision $f(\mathbf{x}; \mathbf{W}, \boldsymbol{\Theta})$ can be written as:

$$f(\mathbf{x}; \mathbf{W}, \boldsymbol{\Theta}) = \sum_{l=0}^{2^h-1} q_l(\mathbf{x}; \mathbf{W}) \boldsymbol{\theta}_l, \text{where} \tag{2}$$

$$q_l(\mathbf{x}; \mathbf{W}) = \prod_{i=0}^{h-1} \sigma\bigg( \big( \langle \mathbf{w}_{i,I(i,l)}, \mathbf{x} \rangle + b_{i,I(i,l)} \big) S(i,l) \bigg).$$

In decision trees, $\sigma(.)$ is the step function, i.e.,

$$\sigma_{\text{step}}(a) = 1, \text{ if } a \geq 0, \text{ and } \sigma_{\text{step}}(a) = 0, \text{ if } a < 0. \tag{3}$$

This hard thresholding $\sigma_{\text{step}}$ presents a key challenge in learning the tree model. A standard approach is to relax $\sigma_{\text{step}}$ via, say, the sigmoid function (Sethi, 1990; Kumar et al., 2017; Biau et al., 2019), and model the tree function as a distribution over the leaf values. Though this readily enables learning via gradient descent-style algorithms, the resulting model is *not* a decision tree as the decision at each node is *soft* thus leading to *exponential* (in height) inference complexity. Also, using standard methods like annealing to convert the soft decisions into hard decisions leads to a significant drop in accuracy (see Table 1).

Applying standard gradient descent algorithm to solve (1) has other potential issues besides the hard/soft decision choices with the choice of $\sigma$. In the following, we leverage three simple but key observations to model the function $f$ in (2), which lets us design efficient and accurate algorithms for solving (1) with any general, potentially unknown, loss function $\ell$.

### 3.1 AND gradients

The $q_l$ function (2) to compute the AND of the decisions along a root-to-leaf path has a multiplicative form, which is commonly used in tree learning formulations (Kumar et al., 2017). When we use the relaxed sigmoid function for $\sigma$, this form implies that the gradient and the Hessian of the function with respect to the parameters are highly ill-conditioned (multiplication of $h$ sigmoids). That is, in certain directions, the Hessian values can be vanishingly small, while others will have a scale of $O(1)$. It is hard to reconcile the disparate scales of the gradient values, leading to poor solutions which we observe empirically as well (in Section 5). We leverage a simple but key observation that $q_l$ can be equivalently written using a sum instead of the multiplicative form:

$$q_l(\mathbf{x}; \mathbf{W}) = \sigma\Big( \sum_{i=0}^{h-1} \sigma\big( \big( \langle \mathbf{w}_{i,I(i,l)}, \mathbf{x} \rangle + b_{i,I(i,l)} \big) S(i,l) \big) - h \Big). \tag{4}$$

**Proposition 1.** *The path function $q$ in (4) is identical to the definition in (2), when $\sigma := \sigma_{step}$ in (3).*

The proposition follows directly from the definition of $\sigma_{\text{step}}$ : $q_l$ term in Eqn. (2) evaluates to 1 iff all terms in the product evaluate to 1 (corresponding to the decisions made along the root-to-leaf path); Similarly in Eqn. (4) (which operates on the same terms as that of $q_l$ in Eqn. (2)) evaluates to 1 iff each term in the sum is 1, because the sum will then be $h$, which after the offset $-h$ yields 1 according to the definition of $\sigma_{\text{step}}$. Therefore the two definitions of $q_l$ are equivalent.

The gradient of $q_l$ in (4) is a sum of $h$ terms each of which is a product of only two sigmoids, so the problem should be much better conditioned. Our hypothesis is validated empirically in Section 5.

### 3.2 Overparameterization

Overparameterization across multiple layers is known to be an effective tool in deep learning with strong empirical and theoretical results (Allen-Zhu et al., 2019; Arora et al., 2019; Sankararaman et al., 2020), and is hypothesized to provide a strong regularization effect. While large height decision trees can seem "deep"

and overparameterized, they do not share some of the advantages of deep overparameterized networks. In fact, deeper trees seem to be harder to train, tend to overfit and provide poor accuracy.

So in this work, we propose overparameterizing by learning "deep" representation of the data point itself, before it is being fed to the decision tree. That is, we introduce $L$ *linear* fully-connected hidden layers with weights $\mathbf{W}^{(1)} \in \mathbb{R}^{d_1 \times d}, \mathbf{W}^{(2)} \in \mathbb{R}^{d_2 \times d_1}, \ldots, \mathbf{W}^{(L-1)} \in \mathbb{R}^{d_{L-1} \times d_{L-2}}, \mathbf{W}^{(L)} \in \mathbb{R}^{2^h - 1 \times d_{L-1}}$, where $d_i$ are hyper-parameters, and apply tree function $f$ to $f(\mathbf{W}^{(L-1)} \mathbf{W}^{(L-2)} \ldots \mathbf{W}^{(1)} \mathbf{x}; \mathbf{W}^{(L)}, \mathbf{\Theta})$. Note that $L$ is the number of hidden layers we use for our overparameterization.

Note that while we introduce many more parameters in the model, due to linearity of the network, each node still has a *linear* decision function: $\langle \mathbf{w}_{ij}^{(L)}, \mathbf{W}^{(L-1)} \mathbf{W}^{(L-2)} \ldots \mathbf{W}^{(1)} \mathbf{x} \rangle = \langle \widetilde{\mathbf{w}}_{ij}, \mathbf{x} \rangle$ where $\widetilde{\mathbf{w}}_{ij} = (\mathbf{W}^{(L-1)} \mathbf{W}^{(L-2)} \ldots \mathbf{W}^{(1)})^\top \mathbf{w}_{ij}^{(L)}$. Thus, we still learn an *oblique decision tree* (Def. 1) and do not require any feature transformation during inference.

**Remark 1.** *We emphasize that despite using overparameterization, we finally obtain standard oblique decision tree (Definition 1) with a linear model at each internal node. In particular, we do not require any feature transformation at the inference time, unlike the tree learning approaches of Kontschieder et al. (2015); Tanno et al. (2019) that learn deep representations thereby increasing the model size and inference time.*

Given that linear overparameterization works with *exactly* the same model class, it is surprising that the addition of linear layers yields significant accuracy improvements in several datasets (Section 5.4). Indeed, there are some theoretical studies (Arora et al., 2018) on why linear networks might accelerate optimization, but we leave a thorough investigation into the surprising effectiveness of linear overparameterization for future work.

### 3.3 Quantization for hard decisions

As mentioned earlier, for training, a common approach is to replace the hard $\sigma_{\text{step}}$ with the *smooth* sigmoid, and then slowly anneal the sigmoid to go towards sharper decisions (Jose et al., 2013; Kumar et al., 2017). That is, start with a small scaling parameter $s$ in sigmoid function $\sigma_s(a) := \frac{1}{1+\exp(-s \cdot a)}$, and then slowly increase the value of $s$. We can try to devise a careful annealing schedule of increasing values of $s$, but the real issue in such a training procedure is that it *cannot* output a hard tree in the end — we still need to resort to heuristics to convert the converged model to a tree, and the resulting loss in accuracy can be steep (see Table 1).

Instead, we use techniques from the quantization-aware neural network literature (Rastegari et al., 2016; Courbariaux et al., 2016; Hubara et al., 2016; Esser et al., 2020), which uses the following trick: apply quantized functions in the forward pass, but use smooth activation function in the backward pass to propagate appropriate gradients. In particular, we leverage the scaled quantizers and the popular "straight-through estimator" to compute the gradient of the hard $\sigma_{\text{step}}$ function (Rastegari et al., 2016; Courbariaux et al., 2016) (discussed in Section 4). Note that, unlike typical quantized neural network scenarios, our setting requires (1-bit) quantization of only the $\sigma$ functions.

Finally, using the aforementioned three ideas, we reformulate the tree learning problem (1) as:

$$\min_{\mathbf{W}^{(m)}, \mathbf{\Theta}} \frac{1}{n} \sum_{i=1}^{n} \ell\big(f(\mathbf{x}_i; \mathbf{W}, \mathbf{\Theta}), y_i\big) + \lambda \Phi_{\text{reg}}(\mathbf{W}, \mathbf{\Theta}), \tag{5}$$

$$\text{s.t.} \quad \mathbf{W} = \mathbf{W}^{(L)} \mathbf{W}^{(L-1)} \ldots \mathbf{W}^{(1)},$$

$$f(\mathbf{x}_i; \mathbf{W}, \mathbf{\Theta}) \text{ as in (2)}, \quad q_l(\mathbf{x}_i; \mathbf{W}) \text{ as in (4)}.$$

## 4 Learning Algorithms

We now present our Dense Gradient Tree, DGT, learning algorithm for solving the proposed decision tree formulation (5) in the bandit setting introduced in Section 2. Extension to the fully supervised (batch) setting using standard mini-batching, called DGT, is presented in Appendix A.

---

**Algorithm 1** DGT-Bandit: Learning decision trees in the bandit setting

---

1: **Input:** height $h$, max rounds $T$, learning rate $\eta$, loss $\ell$, $(\lambda, \Phi_{\text{reg}})$, overparam. $L$, hidden dim $d_i$, $i \in [L]$

2: **Output:** Tree model $\mathbf{W}, \mathbf{\Theta} \in \mathbb{R}^{2^h \times K}$

3: **Init:** $\mathbf{W}_0^{(1)} \in \mathbb{R}^{d_1 \times d}, \mathbf{W}_0^{(2)} \in \mathbb{R}^{d_2 \times d_1}, \ldots, \mathbf{W}_0^{(L)} \in \mathbb{R}^{(2^h-1) \times d_{L-1}}, \mathbf{\Theta}_0 \in \mathbb{R}^{2^h \times K}$ randomly.

4: **for** round $i = 1, 2, \ldots, T$ **do**

5:     Get (unlabeled) example $[\mathbf{x}_i; 1]$   *// with bias*

6:     $\theta_l \leftarrow f(\mathbf{x}_i; \mathbf{W}_{i-1}, \mathbf{\Theta}_{i-1})$ via (2), and $\sigma_{\text{step}}$ in (3)   *// compute the tree prediction*

7:     $\hat{y}_i \leftarrow \begin{cases} \theta_l := \boldsymbol{\theta}_l, & \text{for regression, } K = 1, \text{ by defn.} \\ \text{sampled acc. to (7), } & \text{for classification.} \end{cases}$

8:

9:     $\mathbf{g} \leftarrow \begin{cases} \text{derivative of } \ell \text{ at } \hat{y}_i \text{ via (9), (regression)} \\ \text{gradient computed via (8). (classification)} \end{cases}$   *// bandit feedback*

10:     $\{\nabla_{\mathbf{W}^{(m)}} f, \nabla_{\mathbf{\Theta}} f\} = \text{BackProp}(\mathbf{x}_i; \mathbf{W}_{i-1}, \mathbf{\Theta}_{i-1})$

11:     $\mathbf{W}_i^{(m)} = \mathbf{W}_{i-1}^{(m)} - \mathbf{g} \cdot \eta \nabla_{\mathbf{W}^{(m)}} f - \eta \lambda \nabla_{\mathbf{W}^{(m)}} \Phi_{\text{reg}}(\mathbf{W}, \mathbf{\Theta})$,   for $m = 1, 2, \ldots, L$.

12:     $\mathbf{\Theta}_i = \mathbf{\Theta}_{i-1} - \mathbf{g} \cdot \eta \nabla_{\mathbf{\Theta}} f - \eta \lambda \nabla_{\mathbf{\Theta}} \Phi_{\text{reg}}(\mathbf{W}, \mathbf{\Theta})$

---

**Algorithm 2** BackProp

---

1: **Input:** $\mathbf{x}, \mathbf{W}, \mathbf{\Theta} := \boldsymbol{\theta} \in \mathbb{R}^{2^h}$ $(K = 1)$

2: **Output:** $\nabla_{\mathbf{W}^{(m)}} f(\mathbf{x}; \mathbf{W}, \mathbf{\Theta}), \nabla_{\boldsymbol{\theta}} f(\mathbf{x}; \mathbf{W}, \mathbf{\Theta})$

3: **Gradient with respect to $\mathbf{\Theta}$:**

4:     Let $l^* = \arg\max_l q_l(\mathbf{x}; \mathbf{W})$

5:     $\nabla_{\boldsymbol{\theta}} f(\mathbf{x}; \mathbf{W}, \mathbf{\Theta}) = \mathbf{e}_{l^*}$ (standard basis vector).

6: **Gradient with respect to $\mathbf{W}^{(m)}$, $m = 1, 2, \ldots, L$:**

7:     Let $\mathbf{a} = \mathbf{W}^{(L)}(\ldots(\mathbf{W}^{(2)}(\mathbf{W}^{(1)}\mathbf{x}))) \in \mathbb{R}^{2^h-1}$

8:     Let $a_{i,j}$ denote the entry of $\mathbf{a}$ corresponding to internal node $(i, j)$

9:     Let $\widetilde{q}_l(\mathbf{x}; \mathbf{W}) = \sum_{i=0}^{h-1} \text{sign}(a_{i,I(i,l)}) S(i, l)$

10:     $\nabla_{\mathbf{W}^{(m)}} \widetilde{q}_l(\mathbf{x}; \mathbf{W}) = \sum_{i=0}^{h-1} S(i, l) \mathbb{1}_{|a_{i,I(i,l)}| \leq 1} \nabla_{\mathbf{W}^{(m)}} a_{i,I(i,l)}$   *// Use straight-through estimator for the inner $\sigma$ in (4)*

11:     Let $z = \sum_l \exp\left(\widetilde{q}_l(\mathbf{x}; \mathbf{W})\right)$

12:     Let $v = \sum_l \exp\left(\widetilde{q}_l(\mathbf{x}; \mathbf{W})\right) \theta_l$

13:     $\nabla_{\mathbf{W}^{(m)}} f(\mathbf{x}; \mathbf{W}, \mathbf{\Theta}) = (1/z) \sum_{l=0}^{2^h-1} (\theta_l - v/z) \exp\left(\widetilde{q}_l(\mathbf{x}; \mathbf{W})\right) \nabla_{\mathbf{W}^{(m)}} \widetilde{q}_l(\mathbf{x}; \mathbf{W})$   *// Use SOFTMAX for the outer $\sigma$ in (4).*

---

Recall that in the bandit setting, the learner observes the features $\mathbf{x}_i$ at the $i$th round (but not the label) and a loss (or reward) value for its prediction $\hat{y}_i$. The training procedure, called DGT-Bandit, is presented in Algorithm 1. It is the application of gradient descent to problem (5). The corresponding backpropagation step is presented in Algorithm 2. There are two key challenges in implementing the algorithm: (a) enforcing hard thresholding $\sigma_{\text{step}}$ function discussed in Section 3, and (b) learning the tree parameters with only black-box access to loss $\ell$ and no gradient information.

**(a) Handling $\sigma_{\text{step}}$.** In the forward pass (Line 6 of Algorithm 1), we use $\sigma_{\text{step}}$ defined in (3) to compute the tree prediction. However, during back propagation, we use softer version of the decision function; see Algorithm 2. In particular, we consider the following variant of the $q_l$ function in the backward pass, that applies SOFTMAX for the outer $\sigma$ in (4):

$$\widehat{q}_l(\mathbf{x}; \mathbf{W}) \propto \exp\left(\sum_{i=0}^{h-1} \text{sign}(\langle \mathbf{w}_{i,I(i,l)}, \mathbf{x} \rangle + b_{i,I(i,l)}) S(i, l)\right),$$

where the normalization constant ensures that $\sum_l \widehat{q}_l(\mathbf{x}; \mathbf{W}) = 1$, and $\text{sign}(a) = 2\sigma_{\text{step}}(a) - 1$. By introducing SOFTMAX, we can get rid of the offset $-h$ in (4). To compute the gradient of the sign function, we use the straight-through estimator (Bengio et al., 2013; Hubara et al., 2016) defined as: $\frac{\partial \text{sign}(a)}{\partial a} = \mathbb{1}_{|a| \leq 1}$ (Line 10 of Algorithm 2).

**Remark 2.** *While we use the gradient of the SOFTMAX operator in $\widehat{q}_l$ for updating $\mathbf{W}$ parameters, we use hard* $\arg\max$ *operator for updating $\boldsymbol{\Theta}$ parameters (as in Line 5 of Algorithm 2).*

**(b) Handling bandit feedback.** The problem of learning with bandit feedback is notoriously challenging and has a long line of remarkable ideas (Flaxman et al., 2005; Agarwal et al., 2010). While we can view the problem as a direct online optimization over the parameters $\mathbf{W}, \boldsymbol{\Theta}$, it would lead to poor sample complexity due to dimensionality of the parameter set. Instead, we leverage the fact that the gradient of the prediction $\boldsymbol{\theta}_l = f(\mathbf{x}; \mathbf{W}, \boldsymbol{\Theta})$ with respect to the loss function can be written down as:

$$\nabla_{\mathbf{W}, \boldsymbol{\Theta}} \ell\big(f(\mathbf{x}; \mathbf{W}, \boldsymbol{\Theta})\big) = \ell'(\boldsymbol{\theta}_l) \cdot \nabla_{\mathbf{W}, \boldsymbol{\Theta}} f(\mathbf{x}; \mathbf{W}, \boldsymbol{\Theta}). \tag{6}$$

Now, given $\mathbf{x}$ and $\mathbf{W}, \boldsymbol{\Theta}$, we can compute $\nabla_{\mathbf{W}, \boldsymbol{\Theta}} f(\mathbf{x}; \mathbf{W}, \boldsymbol{\Theta})$ exactly. So only unknown quantity is $\ell'(\boldsymbol{\theta}_l)$ which needs to be estimated by the bandit/point-wise feedback based on the learning setting. Also note that in this case where $\ell'$ is estimated using *one* loss feedback, the gradients are still dense, i.e., all the parameters are updated as $\nabla_{\mathbf{W}, \boldsymbol{\Theta}} f$ is dense; see Lines 11 and 12 of Algorithm 1.

**Classification setting:** This is standard contextual multi-armed bandit setting (Allesiardo et al., 2014), where for a given $\mathbf{x}$, the goal is to find the arm to pull, i.e., output a discrete class for which we obtain a loss/reward value. So, here, $\boldsymbol{\theta}_l \in \mathbb{R}^K$ and the prediction is given by sampling an arm/class $\hat{y} \in [K]$ from the distribution:

$$\hat{y} \sim \mathbf{p}_i, \ \mathbf{p}_i(k) = (1 - \delta)\mathbb{1}[k = \arg\max_{k'} \boldsymbol{\theta}_l(k')] + \delta/K, \tag{7}$$

where $\delta > 0$ is exploration probability. For the prediction $\hat{y}$, we obtain loss $\ell(\hat{y})$. Next, to update the tree model as in (6), we follow Allesiardo et al. (2014) and estimate the gradient $\ell'(\boldsymbol{\theta}_l)$ at $\boldsymbol{\theta}_l$ as:

$$\ell'(\boldsymbol{\theta}_l) = 2 \cdot \mathbf{p}(\hat{y})^{-1} \cdot \big(\ell(\hat{y}) - (1 - \sigma(\boldsymbol{\theta}_l(\hat{y})))\big) \cdot \tag{8}$$
$$\sigma(\boldsymbol{\theta}_l(\hat{y})) \cdot (1 - \sigma(\boldsymbol{\theta}_l(\hat{y}))) \cdot \mathbf{e}_{\hat{y}},$$

with sigmoid $\sigma$ and the $\hat{y}$th basis vector denoted $\mathbf{e}_{\hat{y}}$.

**Regression setting:** Here, the prediction is $\hat{y} = \theta_l := \boldsymbol{\theta}_l$, and $\ell'(\theta_l)$ is a *one-dimensional* quantity which can be estimated using the loss/reward value given for *one* prediction (Flaxman et al., 2005) :

$$\ell'(\hat{y}; \mathbf{x}) \approx \delta^{-1}\ell(\hat{y} + \delta u; \mathbf{x})u, \tag{9}$$

where $u$ is randomly sampled from $\{-1, +1\}$, and $\delta > 0$. Note that even with this simple estimator, Algorithm 1 converges to good solutions with far fewer queries compared to a baseline (Figure 3). In scenarios when there is access to *two-point feedback* (Shamir, 2017), we use (for small $\delta > 0$):

$$\ell'(\hat{y}; \mathbf{x}) \approx (2\delta)^{-1}\big(\ell(\hat{y} + \delta; \mathbf{x}) - \ell(\hat{y} - \delta; \mathbf{x})\big). \tag{10}$$

In our experiments (See Table 6), we find that this estimator performs competitive to the case when $\ell$ is fully known (i.e., gradient is exact).

**Choice of $\Phi_{\mathbf{reg}}$.** It is important to regularize the tree model as (a) it is overparameterized, and (b) we prefer sparse, shallow trees. Note that our model assumes a complete binary tree to begin with, thus regularization becomes key in order to learn sparse parameters in the nodes, which can be pruned if necessary doing a single pass. In our experiments, we use a combination of both $L_1$ and $L_2$ penalty on the weights.

## 5 Experiments

We evaluate our approach on the following aspects:
**1.** Performance on multi-class classification and regression benchmarks, compared to SOTA methods and standard baselines, in **(a)** supervised learning setting, and **(b)** bandit feedback setting.
**2.** Benefits of our design choices (in Section 3) against standard existing techniques.

| Dataset | DGT (Alg. 3) | TAO | LCN | TEL | PAT | Ridge | CART |
|---|---|---|---|---|---|---|---|
| AILERONS | **1.72±0.016** (6) | 1.76±0.02 | 2.21±0.068 | 2.04±0.130 | 2.53±0.050 | 1.75±0.000 | 2.01±0.000 |
| ABALONE | **2.15±0.026** (6) | 2.18±0.05 | 2.34±0.066 | 2.38±0.203 | 2.54±0.113 | 2.23±0.016 | 2.29±0.034 |
| COMP-ACTIV | 2.91±0.149 (6) | **2.71±0.04** | 4.43±0.498 | 5.28±0.837 | 8.36±1.427 | 10.05±0.506 | 3.35±0.221 |
| PDBBIND | 1.39±0.017 (6) | 1.45±0.007 | 1.39±0.017 | 1.53±0.044 | 1.57±0.025 | **1.35±0.000** | 1.55±0.000 |
| CTSLICE | 2.30±0.166 (10) | **1.54±0.05** | 2.18±0.108 | 4.51±0.378 | 12.20±0.987 | 8.29±0.054 | 5.78±0.224 |
| YEARPRED | **9.05±0.012** (8) | 9.11±0.05 | 9.14±0.035 | 9.53±0.079 | 9.90±0.043 | 9.51±0.000 | 9.69±0.000 |
| MICROSOFT | 0.772±0.000 (8) | 0.772±0.000 | - | 0.777±0.003 | 0.797±0.001 | 0.779±0.000 | 0.771±0.000 |
| YAHOO | **0.795±0.001** (8) | 0.796±0.001 | 0.804±0.002 | - | 0.889±0.004 | 0.800±0.000 | 0.807±0.000 |

Table 1: Fully supervised (regression) setting: mean test RMSE ± std. deviation (over 10 runs with different random seeds). Best height is indicated in parentheses for our method DGT (given in Appendix A). For TEL, we set $\gamma = 0.01$.

| Dataset | DGT (Alg. 3) | TAO | LCN | TEL | CART |
|---|---|---|---|---|---|
| PROTEIN | **67.80±0.40** (4) | **68.41±0.27** | 67.52±0.80 | 67.63±0.61 | 57.53±0.00 |
| PENDIGITS | **96.36±0.25** (8) | **96.08±0.34** | 93.26±0.84 | 94.67±1.92 | 89.94±0.34 |
| SEGMENT | **95.86±1.16** (8) | **95.01±0.86** | 92.79±1.35 | 92.10±2.02 | 94.23±0.86 |
| SATIMAGE | **86.64±0.95** (6) | **87.41±0.33** | 84.22±1.13 | 84.65±1.18 | 84.18±0.30 |
| SENSIT | **83.67±0.23** (10) | 82.52±0.15 | 82.02±0.77 | 83.60±0.16 | 78.31±0.00 |
| CONNECT4 | 79.52±0.24 (8) | **81.21±0.25** | 79.71±1.16 | 80.68±0.44 | 74.03±0.60 |
| MNIST | 94.00±0.36 (8) | **95.05±0.16** | 88.90±0.63 | 90.93±1.37 | 85.59±0.06 |
| LETTER | 86.13±0.72 (10) | **87.41±0.41** | 66.34±0.88 | 60.35±3.81 | 70.13±0.08 |
| FORESTCOVER | 79.25±0.50 (10) | **83.27±0.32** | 67.13±3.00 | 73.47±0.83 | 77.85±0.00 |
| CENSUS1990 | 46.21±0.17 (8) | **47.22±0.10** | 44.68±0.45 | 37.95±1.95 | 46.40±0.00 |
| HIV | 0.712±0.020 | 0.627±0.000 | **0.738±0.014** | - | 0.562±0.000 |
| BACE | 0.767±0.045 | 0.734±0.000 | **0.791±0.019** | 0.810±0.028 | 0.697±0.000 |

Table 2: Fully supervised setting: Comparison of tree learning methods on various **classification** datasets. Results are computed over 10 runs with different random seeds. Numbers in the first 10 rows are mean test accuracy (%) ± std. deviation, and the last 2 rows are mean test AUC ± std. deviation. For TEL, we set $\gamma = 0.01$.

**Datasets.** We use all regression datasets from three recent works: 3 large tabular datasets from Popov et al. (2020), a chemical dataset from Lee and Jaakkola (2020) and 4 standard (UCI) datasets from Zharmagambetov and Carreira-Perpinan (2020). For classification, we use 8 multi-class datasets from Carreira-Perpinán and Tavallali (2018) and 2 large multi-class datasets from Féraud et al. (2016) on which we report accuracy and 2 binary chemical datasets from Lee and Jaakkola (2020) where we follow their scheme and report AUC. Sizes, splits and other details for all the datasets are given in Appendix B.

**Implementation details.** We implemented our algorithms in PyTorch v1.7 with CUDA v11[2]. We experimented with different quantizers (Rastegari et al., 2016; Courbariaux et al., 2016; Esser et al., 2020) for $\sigma_{\text{step}}$ in (4), but they did not yield any substantial improvements, so we report results for the implementation as given in Algorithm 2. For LCN (Lee and Jaakkola, 2020), TEL (Hazimeh et al., 2020), and CBF (Féraud et al., 2016), we use their publicly available implementations (LCN; TEL; CBF). For TAO (Zharmagambetov and Carreira-Perpinan, 2020; Carreira-Perpinán and Tavallali, 2018), we did our best to implement their algorithms in Python (using liblinear solver in scikit-learn) since the authors have not made their code available yet (Zharmagambetov, April 2021). For (linear) contextual-bandit algorithm with $\epsilon$-greedy exploration (Eqn. (6) and Algorithm 2 in Bietti et al. (2018)), we use VowpalWabbit (VW) with the --cb_explore and --epsilon flags. We train CART and Ridge regression using scikit-learn. We tuned relevant hyper-parameters on validation sets (details in Appendix C.3). For all tree methods we vary $h \in \{2, 4, 6, 8, 10\}$, (1) for LCN: optimizer, learning-rate, dropout, and hidden layers of $g_\phi$; (2) for TAO: $\lambda_1$ regularization; (3) for DGT: momentum, regularization $\lambda_1, \lambda_2$ and overparameterization $L$, (4) for TEL: learning rate and regularization $\lambda_2$.

We report all metrics averaged over ten random runs, and statistical significance based on unpaired t-test at a significance level of 0.05.

## 5.1 Supervised (tree) learning setting

In this setting, both the loss $\ell$ as well as label information is given to the learner.

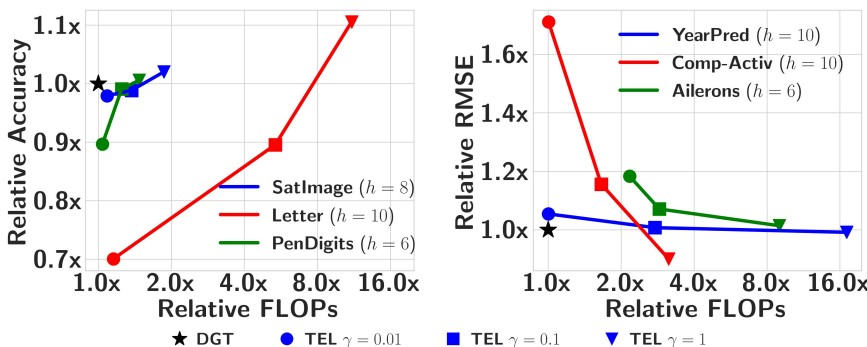

Figure 1: Performance vs. inference FLOPS (log scale) of the TEL method (for different $\gamma$ values that controls the number of leaves reached), relative to DGT, shown for the best performing heights for TEL on the respective datasets.

### 5.1.1 Regression

The goal is to a learn decision tree that minimizes the root mean squared error (RMSE) between the target and the predictions. Test RMSEs of the methods (for the best-performing heights) are given in Table 1. We consider three SOTA tree methods: (1) TEL, with $\gamma = 0.01$ to ensure learning hard decision trees, (2) TAO (3) LCN, and baselines: (4) (axis-parallel) CART, (5) probabilistic tree with annealing (PAT), discussed in the beginning of Section 3.3, and (6) linear regression ("Ridge"). We report numbers from the TAO paper where available. DGT (with mini-batching in Appendix A) performs better than both Ridge and CART baselines on almost all the datasets. On 3 datasets, DGT has lower RMSE (statistically significant) than TAO; on 2 datasets (COMP-ACTIV, CTSLICE), TAO outperforms ours, and on the remaining 3, they are statistically indistinguishable. We clearly outperform LCN on 4 out of 6 datasets, and are competitive on 1. [3]

### 5.1.2 Classification

Here, we wish to learn a decision tree that minimizes misclassification error by training using cross-entropy loss. Test performance (accuracy or AUC as appropriate) of the methods (for the best-performing heights) are presented in Table 2. We compare against the same set of methods as used in regression, excluding PAT and linear regression. Of the 12 datasets studied, DGT achieves statistically significant improvement in performance on all but 1 dataset against CART, on 7 against LCN, and on 3 against TAO. Against TAO, which is a SOTA method, our method performs better on 3 datasets, worse on 5 datasets and on 4 datasets the difference is not statistically significant.

The non-greedy tree learning work of Norouzi et al. (2015) reported results on 7 out of 12 datasets in their paper. What we could deduce from their figures (we do not have exact numbers) is that their method outperforms all our compared methods in Table 2 only on the LETTER dataset (where they achieve $\sim 91\%$ accuracy); on others, they are similar or worse. Similarly, the end-to-end tree learning work of Hehn et al. (2019) reports results on 4 out of 12 classification datasets. Again, deducing from their figures, we find their model performs similarly or worse on the 4 datasets compared to DGT.

### 5.2 Comparison to soft trees, ensembles and trees with linear leaves

**Soft/Probabilistic trees.** DGT learns hard decision trees, with at most $d \cdot h$ FLOPS[4] per inference, as against soft/probabilistic trees with exponential (in $h$) inference FLOPS (e.g., PAT method of Table 1 *without* annealing). In Tables 1 and 2, we note that state-of-the-art TEL method, with $\gamma = 0.01$ (to ensure that at most 1 or 2 leaves are reached per example), performs significantly worse than DGT in almost all the datasets. We now look at the trade-off between computational cost and performance of TEL, that uses a carefully-designed activation function for predicates, instead of the standard sigmoid used in probabilistic trees. The hyper-parameter $\gamma$ in TEL controls the *softness* of the tree, affecting the number of leaves an

---

[3]Despite our efforts, LCN does not converge on MICROSOFT dataset, while TEL doesn't converge on YAHOO and HIV (within the cut-off period of 48 hours)

[4]FLOPS: floating-point operations

| Dataset | DGT | TEL ($\gamma = 1$) | TEL ($\gamma = 0.1$) | TEL ($\gamma = 0.01$) |
|---|---|---|---|---|
| PENDIGITS | $96.36 \pm 0.25$ | $96.54 \pm 0.46$ | $95.10 \pm 1.23$ | $86.09 \pm 2.53$ |
| SATIMAGE | $86.64 \pm 0.95$ | $87.64 \pm 1.23$ | $84.92 \pm 0.94$ | $84.12 \pm 1.20$ |
| LETTER | $86.13 \pm 0.72$ | $95.21 \pm 0.22$ | $77.14 \pm 0.98$ | $60.35 \pm 3.81$ |
| YEARPRED | $9.05 \pm 0.012$ | $8.99 \pm 0.067$ | $9.14 \pm 0.055$ | $9.56 \pm 0.081$ |
| AILERONS | $1.72 \pm 0.016$ | $1.75 \pm 0.062$ | $1.85 \pm 0.089$ | $2.05 \pm 0.013$ |
| COMP-ACTIV | $2.91 \pm 0.149$ | $2.80 \pm 0.205$ | $3.60 \pm 0.564$ | $5.12 \pm 0.863$ |

Table 3: Performance comparison with soft trees on various classification and regression datasets. Results are computed over 10 runs with different random seeds. Numbers in the first 3 rows are mean test accuracy (%) $\pm$ std. deviation, and the last three rows are mean test RMSE $\pm$ std. deviation. For TEL we show performance with different settings of $\gamma$ leading to different degrees of *softness*.

| Dataset | DGT-Forest | XGBoost | AdaBoost |
|---|---|---|---|
| AILERONS | $\mathbf{1.65 \pm 0.00 \ (6)}$ | $1.72 \pm 0.00 \ (7)$ | $1.75 \pm 0.00 \ (15)$ |
| ABALONE | $\mathbf{2.08 \pm 0.00 \ (8)}$ | $2.20 \pm 0.00 \ (10)$ | $2.15 \pm 0.00 \ (10)$ |
| COMP-ACTIV | $2.63 \pm 0.08 \ (8)$ | $2.57 \pm 0.00 \ (10)$ | $\mathbf{2.56 \pm 0.11 \ (10)}$ |
| CTSLICE | $1.22 \pm 0.07 \ (10)$ | $\mathbf{1.18 \pm 0.00 \ (10)}$ | $1.31 \pm 0.01 \ (10)$ |
| YEARPRED | $\mathbf{8.91 \pm 0.00 \ (8)}$ | $9.01 \pm 0.00 \ (10)$ | $9.21 \pm 0.03 \ (15)$ |

Table 4: Supervised tree ensembles: mean test RMSE $\pm$ std. deviation (over 10 runs with different random seeds) for forest methods on regression datasets. DGT-Forest uses 30 trees while XGBoost and AdaBoost use about 1000 trees. Maximum height of the trees is given in parentheses.

example lands in, and in turn, the inference FLOPS. In Figure 1, we show the relative performance (i.e. ratio of accuracy or RMSE of TEL to that of DGT for a given height) vs relative FLOPS (i.e. ratio of mean inference FLOPS of TEL to that of DGT for a given height) of TEL, for different $\gamma$ values. First, note that, by definition, DGT is at (1,1) in the plots. When $\gamma = 0.01$, TEL achieves about $d \cdot h$ inference FLOPS on average, similar to that of a hard decision tree (as in Table 1), but its performance is significantly worse than DGT on all the datasets. On the other hand, TEL frequently outperforms DGT at $\gamma = 1$ (for instance, on SATIMAGE, it improves accuracy by 2%), but at a significant computational cost (as much as 17x FLOPS over DGT)[5]. We also present the performance comparison between DGT and TEL (at different $\gamma$ settings) in Table 3. Overall, DGT achieves competitive performance while keeping the inference cost minimal.

**Tree ensembles.** To put the results shown so far in perspective, we compare DGT to widely-used ensemble methods for supervised learning. To this end, we extend DGT to learn forests (DGT-Forest) using the bagging technique, where we train a fixed number of DGT tree models independently on a bootstrap sample of the training data and aggregate the predictions of individual models (via voting or averaging) to generate a prediction for the forest. We compare DGT-Forest with AdaBoost and XGBoost in Table 4. First, as expected, we see that DGT-Forest outperforms DGT (in Table 1) on all datasets. Next, DGT-Forest using only 30 trees outperforms the two tree ensemble methods that use as many as 1000 trees, on 3 out of 5 datasets, and is competitive in the other 2. We also find that our results improve over other standard ensemble methods, on the same datasets and splits, published in Zharmagambetov and Carreira-Perpinan (2020).

**Trees with linear leaves.** While DGT produces oblique trees whose leaves contain a constant model (a learnt constant value), we also compare it with trees containing a linear model in the leaves. In general, a tree with linear leaves is more expressive than a tree of the same height with constant leaves.[6] In Table 5 we present a comparison with such trees learnt by TAO. As expected, we see that these trees usually outperform those learnt by DGT.

---

[5]TEL is still much better than standard probabilistic trees with $\sim$ 100x FLOPS for $h = 10$.

[6]In the case of classification, a tree of height $h$ with linear leaves can be equivalently represented by a tree of height $O(K) + h$, where $K$ is the number of classes. For regression though, such equivalence does not hold.

| Dataset | DGT (with constant leaves) | TAO (with linear leaves) |
|---|---|---|
| AILERONS | $1.72 \pm 0.016$ | $1.74 \pm 0.01$ |
| ABALONE | $2.15 \pm 0.026$ | $2.07 \pm 0.01$ |
| COMP-ACTIV | $2.91 \pm 0.149$ | $2.58 \pm 0.02$ |
| CTSLICE | $2.30 \pm 0.166$ | $1.16 \pm 0.02$ |
| YEARPRED | $9.05 \pm 0.012$ | $9.08 \pm 0.03$ |

Table 5: Comparing DGT trees with constant leaves and TAO trees with linear model in the leaves. Scores are mean test RMSE $\pm$ std. deviation (over 10 runs for DGT and over 5 runs for TAO as reported in Zharmagambetov and Carreira-Perpinan (2020)).

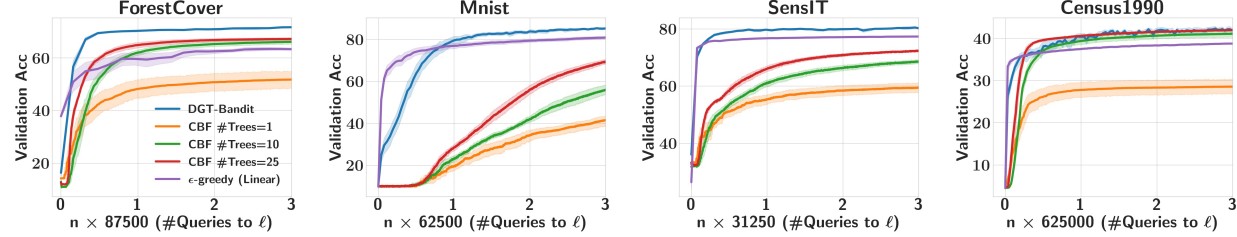

Figure 2: Bandit feedback setting (**classification**, one-point estimator in (8): #Queries to $\ell$ vs mean accuracy on held-out set for multi-class datasets over 10 runs, with 95% confidence intervals. For DGT-Bandit, we use $h = 6$ for all datasets except CENSUS1990 where we use $h = 8$.

## 5.3 Bandit feedback setting

In the bandit setting, given a data point, we provide prediction using the tree function, and based on the loss for the prediction, we update the tree function. In particular, in this setting, neither the loss function nor the label information is available for the learner.

### 5.3.1 Classification

This is the standard contextual multi-armed bandit setting. Here, we compare with CBF (Féraud et al., 2016) and the $\epsilon$-greedy contextual bandits algorithm (Bietti et al., 2018) with linear policy class. Many SOTA tree-learning methods (Carreira-Perpinán and Tavallali, 2018; Zharmagambetov and Carreira-Perpinan, 2020; Bertsimas et al., 2019) do not apply to the bandit setting, as their optimization requires access to full loss function, so as to evaluate/update node parameters along different paths. For instance, standard extension of TAO would require evaluation of the loss function for a given point on $O(2^h)$ predictions. Similarly, directly extending non-greedy method of Norouzi et al. (2015) would require $h + 1$ queries to the loss $\ell$ *on the same example* per round, compared to DGT that can work with *one* loss function evaluation.

**Results.** We are interested in the sample complexity of the methods, i.e., *how many queries to the loss $\ell$ is needed for the regret to become very small*. In Figure 2, we show the convergence of accuracy against the #queries ($n$) to $\ell$ (0-1 loss) for DGT-Bandit (Algorithm 1), CBF (because their trees are axis-aligned, we use forests), and the $\epsilon$-greedy baseline, on the large multi-class classification datasets. Each point on the curve is the (mean) multi-class classification accuracy (and the shaded region corresponds to 95% confidence interval) computed on a *fixed held-out* set for the (tree) model obtained after $n$ rounds. It is clear that our method takes far fewer examples to converge to a fairly accurate solution in all the datasets shown. On FORESTCOVER with 7 classes, after 10,000 queries (2% of train), our method achieves a test accuracy of 62.5% which is only 16% worse compared to the best solution achieved using full supervision. On MNIST, CBF (with 25 trees) takes nearly 400K queries ($\sim$6.7 passes over train) to reach an accuracy of $\sim$80%, which DGT-Bandit reaches within 76K queries ($\sim$1.25 passes).

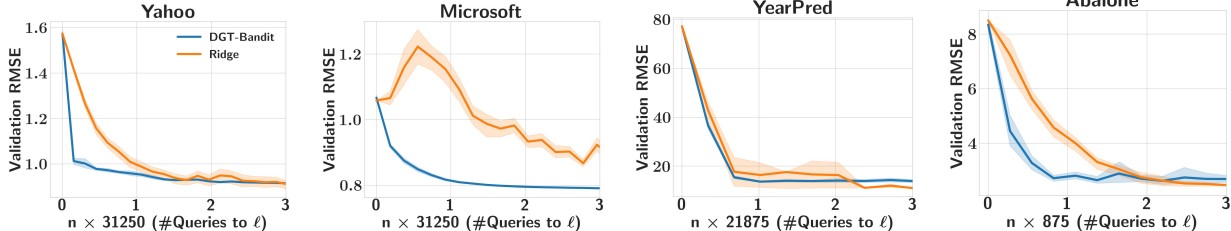

Figure 3: Bandit setting (**regression**, one-point estimator in (9)): #Queries to $\ell$ vs RMSE on held-out set for regression datasets, averaged over 10 runs and 95% confidence intervals. Our method is run with default hyperparameters and $h = 8$.

| Dataset | DGT-Bandit (Alg. 1) | | Linear Regression | |
|---|---|---|---|---|
| | Squared | Huber | Squared | Huber |
| AILERONS | **1.71±0.016** | **1.72±0.016** | 1.78±0.003 | 1.87±0.001 |
| ABALONE | **2.16±0.030** | **2.16±0.029** | 2.28±0.016 | 2.44±0.078 |
| COMP-ACTIV | **3.05±0.199** | **2.99±0.166** | 10.05±0.412 | 10.27±0.202 |
| PDBBIND | 1.39±0.022 | 1.40±0.023 | **1.35±0.006** | **1.36±0.001** |
| CTSLICE | **2.36±0.212** | **2.42±0.168** | 8.32±0.051 | 8.52±0.042 |
| YEARPRED | **9.05±0.013** | **9.06±0.008** | 9.54±0.002 | 9.68±0.003 |
| MICROSOFT | **0.772±0.000** | **0.773±0.001** | 0.781±0.000 | 0.782±0.000 |
| YAHOO | **0.796±0.001** | **0.797±0.002** | 0.810±0.001 | 0.815±0.000 |

Table 6: Bandit setting (**regression**): Mean test RMSE ± std. deviation (over 10 runs with different random seeds) using two-point estimator for $\ell'$ in (10).

### 5.3.2 Regression

We evaluate our method in this setting with two different losses (unknown to the learner): (a) squared loss $\ell_{\text{sq}}(\hat{y}, y) = (\hat{y} - y)^2$, and (b) the Huber loss defined as:

$$\ell_{\xi}(\hat{y}, y) = \begin{cases} \dfrac{1}{2}(\hat{y} - y)^2, & \text{if } |\hat{y} - y| \leq \xi, \\ \xi|\hat{y} - y| - \dfrac{1}{2}\xi^2, & \text{otherwise.} \end{cases}$$

We use ridge regression with bandit feedback as a baseline (which is as good as any tree method in the fully supervised setting, Table 1, on 4 out of 8 datasets) in this setting, for which gradient estimation is well known (using perturbation techniques described in Section 4).

**Results.** We are interested in the sample complexity of the methods, i.e., *how many queries to the loss $\ell$ is needed for the regret to become very small*. In Figure 3, we show the convergence of RMSE against the # queries ($n$) to loss $\ell_{sq}$ for our method DGT-Bandit and the baseline (online) linear regression, both implemented using the one-point estimator for the derivative $\ell'$ in (9). Each point on the curve is the RMSE computed on a *fixed held-out* set for the (tree) model obtained after $n$ rounds (in Algorithm 1). Our solution takes far fewer examples to converge to a fairly accurate solution in all the datasets shown. For instance, on the YAHOO dataset after 60,000 queries (13% of training data), our method achieves a test RMSE of 0.858 which is only 7% worse compared to the best solution achieved using full supervision (see Table 1).

In Table 6, we show a comparison of the methods implemented using the two-point estimator in (10), on all the regression datasets, for two loss functions. Here, we present the final test RMSE achieved by the methods after convergence. Not surprisingly, we find that our solution indeed does consistently better on most of the datasets. What is particularly striking is that, on many datasets, the test RMSE nearly matches the corresponding numbers (in Table 1) in the fully supervised setting where the exact gradient is known!

### 5.4 Ablative Analysis

We empirically validate the key design choices of DGT motivated in Section 3. In all the cases, we vary *only* the aspect under study, fix all other implementation aspects to our proposed modeling choices, and report

| Dataset | (a) Sum vs Prod. form $q_l$ | | (b) Overparameterization | | (c) Quantization vs Anneal | |
|---|---|---|---|---|---|---|
| | Prod, Eqn. (2) | Sum, Eqn. (4) | $L = 1$ | $L = 3$ | $\sigma_s$ + Anneal | sign |
| AILERONS | 2.02±0.028 | 1.72±0.016 | 1.90±0.026 | 1.72±0.016 | 1.73±0.021 | 1.72±0.016 |
| ABALONE | 2.40±0.064 | 2.15±0.026 | 2.21±0.030 | 2.15±0.026 | 2.31±0.06 | 2.15±0.026 |
| YEARPRED | 9.72±0.063 | 9.05±0.012 | 9.18±0.017 | 9.05±0.012 | 9.18±0.04 | 9.05±0.012 |

Table 7: Ablation study of design choices in DGT(mean test RMSE ± std. deviation over 10 random seeds)

the performances of the best (cross-validated) models.

**1. Sum vs Product forms for paths.** In our formulation, we use the sum form of path function $q_l$ as given in (4). We also evaluate the multiplicative form in (2) (used in PAT). Table 7 (a) shows that the product form has consistently worse (e.g., over 17% in AILERONS) RMSE across the datasets than the ones trained with the sum form in our method. This indicates that our hypothesis about gradient ill-conditioning for the product form holds true.

**2. Effect of Overparameterization.** Next, we study the benefit of learning a "deep embedding" for data points via multiple, *linear*, fully-connected layers as described in Section 3.2. It is not clear a priori why such "spurious" linear layers can help at all, as the hypothesis space remains the same. But, we find supportive evidence from Table 7 (b) that adding just 2 layers helps improve the performance in many datasets. On YEARPRED, overparameterization makes the difference between SOTA methods and ours. Recent findings (Arora et al., 2018) suggest that linear overparameterization helps optimization via acceleration. We find that a modest value of $L = 3$ works across datasets. We defer a thorough study of this aspect and its implications to future work.

**3. Quantization vs Annealing.** Finally, we study the effectiveness of the quantized gradient updates, i.e., using the sign function in the forward pass, and the straight-through estimator to compute its gradient in the backward pass. We compare against the annealing technique for scaled sigmoid function $\sigma_s(a) = \frac{1}{1+\exp(-s \cdot a)}$, i.e., slowly increasing the value of $s$ during training. Table 7 (c) suggests that the quantized updates are crucial for DGT; on ABALONE, annealing performs worse than all the methods in Table 1, except PAT which it improves over because of overparameterization and the use of sum form.

## 6 Conclusions

We proposed an end-to-end gradient-based method for learning hard decision trees that admits dense updates to all tree parameters. DGT enables learning trees in several practical scenarios, such as supervised learning and online learning with limited feedback. Our comprehensive experiments in the supervised setting demonstrated that DGT achieves nearly SOTA performance on several datasets. On the other hand, in the bandit setting, where most existing techniques do not even apply, DGT yields accurate solutions with low sample complexity, and in many cases, matches the performance of our offline-trained models. In our experiments, default hyperparameters worked well (see Appendix C.3), but it is unclear how to set them in real-world online deployments where dataset characteristics might be significantly different. We plan to explore relevant approaches from parameter-free online learning literature (Chap. 9, Orabona (2019)). Understanding linear overparameterization, extending quantization techniques to learning trees with higher out-degree, and deploying DGT in ML-driven systems that need accurate, low cost models, are potential directions at this point.

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

## A   Tree Learning Algorithm Variants

Our method DGT for learning decision trees in the standard supervised learning setting is given in Algorithm 3. Note that here we know the loss function $\ell$ exactly, so computing the gradient is straight-forward (in Step 9). Also, the dot product in Step 9 (as well as in Steps 11 and 12 of Algorithm 1) is appropriately defined when $K > 1$, as it involves a vector and a tensor. The back propagation procedure, which is used both in DGT and in DGT-Bandit (Algorithm 1), is given in Algorithm 2 ($K = 1$ case, for clarity).

---

**Algorithm 3** DGT: Learning decision trees in the supervised learning (batch) setting

---

1: **Input:**   training data $(\mathbf{x}_i, y_i)_{i=1}^n$, height $h$, max epochs $\tau_{\max}$, learning rate $\eta$, loss $\ell$, regularization $(\lambda, \Phi_{\mathrm{reg}})$, overparameterization $L$, hidden dim $d_i$, $i \in [L]$

2: **Output:**   Tree model: $\mathbf{W}$, and $\boldsymbol{\Theta} \in \mathbb{R}^{2^h \times K}$

3: **Init:** $\mathbf{W}_0^{(1)} \in \mathbb{R}^{d_1 \times d}, \mathbf{W}_0^{(2)} \in \mathbb{R}^{d_2 \times d_1}, \ldots, \mathbf{W}_0^{(L)} \in \mathbb{R}^{(2^h - 1) \times d_{L-1}}, \boldsymbol{\Theta}_0$ randomly.

4: $\mathbf{x}_i = [\mathbf{x}_i;\ 1]$ for $i = 1, 2, \ldots, n$  *// incorporate bias*

5: **for** epoch $1 \le \tau \le \tau_{\max}$ **do**

6:    $(\mathbf{W}, \boldsymbol{\Theta}) = (\mathbf{W}_{\tau-1}, \boldsymbol{\Theta}_{\tau-1})$

7:    **for** each mini-batch $B$ **do**  *// perform SGD on mini-batches*

8:       $\boldsymbol{\theta}_l = f(\mathbf{x}_i; \mathbf{W}, \boldsymbol{\Theta}), i \in B$ (via (2), and hard $\sigma$ in (3))  *// Compute the tree model prediction*

9:       $\{\nabla_{\mathbf{W}^{(m)}} \ell, \nabla_{\boldsymbol{\Theta}} \ell\} \leftarrow \frac{1}{|B|} \sum_{i \in B} \nabla_{\boldsymbol{\theta}_l} \ell(y_i, \boldsymbol{\theta}_l) \cdot \mathrm{BackProp}(\mathbf{x}_i; \mathbf{W}, \boldsymbol{\Theta})$  *// Algorithm 2*

10:       $\mathbf{W}^{(m)} \leftarrow \mathbf{W}^{(m)} - \eta \nabla_{\mathbf{W}^{(m)}} \ell - \eta \lambda \nabla_{\mathbf{W}^{(m)}} \Phi_{\mathrm{reg}}(\mathbf{W}, \boldsymbol{\Theta}),$  for $m = 1, 2, \ldots, L$.

11:       $\boldsymbol{\Theta} \leftarrow \boldsymbol{\Theta} - \eta \nabla_{\boldsymbol{\Theta}} \ell - \eta \lambda \nabla_{\boldsymbol{\Theta}} \Phi_{\mathrm{reg}}(\mathbf{W}, \boldsymbol{\Theta})$

12:    $(\mathbf{W}_\tau, \boldsymbol{\Theta}_\tau) = (\mathbf{W}, \boldsymbol{\Theta})$

13: **return** $\mathbf{W}_{\tau_{\max}}, \boldsymbol{\Theta}_{\tau_{\max}}$

---

## B   Datasets Description

Whenever possible we use the train-test split provided with the dataset. In cases where a separate validation split is not provided, we split training dataset in $0.8 : 0.2$ proportion to create a validation. When no splits are provided, we create five random splits of the dataset and report the mean test scores across those splits, following Zharmagambetov and Carreira-Perpinan (2020). Unless otherwise specified, we normalize the input features to have mean 0 and standard deviation 1 for all datasets. Additionally, for regression datasets, we min-max normalize the target to $[0, 1]$.

### B.1   Regression datasets

We used eight scalar regression datasets comprised of the union of datasets used in  Popov et al. (2020); Lee and Jaakkola (2020); Zharmagambetov and Carreira-Perpinan (2020). Table 9 summarizes the dataset details.

- AILERONS : Given attributes describing the status of the aircraft, predict the command given to its ailerons.

- ABALONE : Given attributes describing physical measurements, predict the age of an abalone. We encode the categorical ("sex") attribute as one-hot.

- COMP-ACTIV : Given different system measurements, predict the portion of time that CPUs run in user mode. Copyright notice can be found here: http://www.cs.toronto.edu/ delve/copyright.html.

- CTSLICE : Given attributes as histogram features (in polar space) of the Computer Tomography (CT) slice, predict the relative location of the image on the axial axis (in the range [0 180]).

- PDBBIND : Given standard "grid features" (fingerprints of pairs between ligand and and protein; see Wu et al. (2018)), predict binding affinities. MIT License (https://github.com/deepchem/deepchem/blob/master/LICENSE).

- YEARPRED : Given several song statistics/metadata (timbre average, timbre covariance, etc.), predict the age of the song. This is a subset of the UCI Million Songs dataset.

- MICROSOFT : Given 136-dimensional feature vectors extracted from query-url pairs, predict the relevance judgment labels which take values from 0 (irrelevant) to 4 (perfectly relevant).

- YAHOO : Given 699-dimensional feature vectors from query-url pairs, predict relevance values from 0 to 4.

## B.2 Classification datasets

We use eight multi-class classification datasets from Carreira-Perpinán and Tavallali (2018), two multi-class classification datasets from Féraud et al. (2016) and two binary classification datasets from from Lee and Jaakkola (2020). Table 8 summarizes the dataset details.

- PROTEIN : Given features extracted from amino acid sequences, predict secondary structure for a protein sequence.

- PENDIGITS : Given integer attributes of pen-based handwritten digits, predict the digit.

- SEGMENT : Given 19 attributes for each 3x3 pixel grid of instances drawn randomly from a database of 7 outdoor colour images, predict segmentation of the central pixel.

- SATIMAGE : Given multi-spectral values of pixels in 3x3 neighbourhoods in a satellite image, predict the central pixel in each neighbourhood.

- SENSIT : Given 100 relevant sensor features, classify the vehicles.

- CONNECT4 : Given legal positions in a game of connect-4, predict the game theoretical outcome for the first player.

- MNIST : Given 784 pixels in handwritten digit images, predict the actual digit.

- LETTER : Given 16 attributes obtained from stimulus observed from handwritten letters, classify the actual letters.

- FORESTCOVER : Given 54 cartographic variables of wilderness areas, classify forest cover type.

- CENSUS1990 : Given 68 attributes obtained from US 1990 census, classify the *Yearsch* column.

- HIV : Given standard Morgan fingerprints (2,048 binary indicators of chemical substructures), predict HIV replication. MIT License (https://github.com/deepchem/deepchem/blob/master/LICENSE).

- BACE : Given standard Morgan fingerprints (2,048 binary indicators of chemical substructures), predict binding results for a set of inhibitors. MIT License (https://github.com/deepchem/deepchem/blob/master/LICENSE).

## C Evaluation Details and Additional Results

All scores are computed over 10 runs, differing in the random initialization done, except for TAO on regression datasets which uses 5 runs. In all the experiments, for a fixed tree height and a dataset, our DGT implementation takes $\leq 20$ minutes for training (and $\leq 1$ hour including hyper-parameter search over four GPUs.)

Table 8: Classification datasets description

| dataset | # features | # classes | train size | splits (tr:val:test) | # shuffles | source |
|---|---|---|---|---|---|---|
| CONNECT4 | 126 | 3 | 43,236 | 0.64 : 0.16 : 0.2 | 1 | LIBSVM[7] |
| MNIST | 780 | 10 | 48,000 | Default[8] | 1 | LIBSVM[2] |
| PROTEIN | 357 | 3 | 14,895 | Default[3] | 1 | LIBSVM[2] |
| SENSIT | 100 | 3 | 63,058 | Default[5] | 1 | LIBSVM[2] |
| LETTER | 16 | 26 | 10,500 | Defaul[3] | 1 | LIBSVM[2] |
| PENDIGITS | 16 | 10 | 5,995 | Default[3] | 1 | LIBSVM[2] |
| SATIMAGE | 36 | 6 | 3,104 | Default[3] | 1 | LIBSVM[2] |
| SEGMENT | 19 | 7 | 1,478 | 0.64 : 0.16 : 0.2 | 1 | LIBSVM[2] |
| FORESTCOVER | 54 | 7 | 371,846 | 0.64 : 0.16 : 0.2 | 1 | UCI[8] |
| CENSUS1990 | 68 | 18 | 1,573,301 | 0.64 : 0.16 : 0.2 | 1 | UCI[8] |
| BACE | 2,048 | 2 | 1,210 | Default[3] | 1 | MoleculeNet[9] |
| HIV | 2,048 | 2 | 32,901 | Default[3] | 1 | MoleculeNet[4] |

Table 9: Regression datasets description

| dataset | # features | train size | splits (tr:val:test) | # shuffles | source |
|---|---|---|---|---|---|
| AILERONS | 40 | 5,723 | Default[3] | 1 | LIACC[10] |
| ABALONE | 10 | 2,004 | 0.5 : 0.1 : 0.4 | 5 | UCI[8,11] |
| COMP-ACTIV | 21 | 3,932 | 0.5 : 0.1 : 0.4 | 5 | Delve[12,6] |
| CTSLICE | 384 | 34,240 | 0.5 : 0.1 : 0.4 | 5 | UCI[8,6] |
| PDBBIND | 2,052 | 9,013 | Default[3] | 1 | MoleculeNet[4] |
| YEARPRED | 90 | 370,972 | Default[3] | 1 | UCI[13] |
| MICROSOFT | 136 | 578,729 | Default[3] | 1 | MSLR-WEB10K[14] |
| YAHOO | 699 | 473,134 | Default[3] | 1 | Yahoo Music Ratings[15] |

## C.1 Height-wise results for the fully supervised setting

Heightwise results for our method DGT and compared methods on various regression and classification datasets can be found in Fig. 4 and Fig. 5 respectively. As mentioned in Section 5.1, we find that the

| Dataset | DGT (Alg. 3) | TAO | LCN | TEL | CART |
|---|---|---|---|---|---|
| PROTEIN | **67.80±0.40** (4) | **68.41±0.27** | 67.52±0.80 | 67.63±0.61 | 57.53±0.00 |
| PENDIGITS | **96.36±0.25** (8) | **96.08±0.34** | 93.26±0.84 | 94.67±1.92 | 89.94±0.34 |
| SEGMENT | **95.86±1.16** (8) | **95.01±0.86** | 92.79±1.35 | 92.10±2.02 | 94.23±0.86 |
| SATIMAGE | **86.64±0.95** (6) | **87.41±0.33** | 84.22±1.13 | 84.65±1.18 | 84.18±0.30 |
| SENSIT | **83.67±0.23** (10) | 82.52±0.15 | 82.02±0.77 | 83.60±0.16 | 78.31±0.00 |
| CONNECT4 | 79.52±0.24 (8) | **81.21±0.25** | 79.71±1.16 | 80.68±0.44 | 74.03±0.60 |
| MNIST | 94.00±0.36 (8) | **95.05±0.16** | 88.90±0.63 | 90.93±1.37 | 85.59±0.06 |
| LETTER | 86.13±0.72 (10) | **87.41±0.41** | 66.34±0.88 | 60.35±3.81 | 70.13±0.08 |
| FORESTCOVER | 79.25±0.50 (10) | **83.27±0.32** | 67.13±3.00 | 73.47±0.83 | 77.85±0.00 |
| CENSUS1990 | 46.21±0.17 (8) | **47.22±0.10** | 44.68±0.45 | 37.95±1.95 | 46.40±0.00 |
| HIV | 0.712±0.020 | 0.627±0.000 | **0.738±0.014** | - | 0.562±0.000 |
| BACE | 0.767±0.045 | 0.734±0.000 | **0.791±0.019** | 0.810±0.028 | 0.697±0.000 |

Table 10: Fully supervised setting: Comparison of tree learning methods on various **classification** datasets. Results are computed over 10 runs with different random seeds. Numbers in the first 10 rows are mean test accuracy (%) ± std. deviation, and the last 2 rows are mean test AUC ± std. deviation. For TEL, we set $\gamma = 0.01$.

numbers obtained from our implementation of TAO (Zharmagambetov and Carreira-Perpinan, 2020) differ from their reported numbers on some datasets; so we quote numbers directly from their paper in Table 1, where available. Keeping up with that, in Figure 4, we present height-wise results for the TAO method only on the 3 datasets that they do not report results on. We do not present heightwise results for TEL as we find that the method is generally poor across heights when we restrict $\gamma$ to be small ($\sim 0.01$).

## C.2 Sparsity of learned tree models

Though we learn complete binary trees, we find that the added regularization renders a lot of the nodes in the learned trees expendable, and therefore can be pruned. For instance, out of 63 internal nodes in a height 6 AILERONS tree (best model), only 47 nodes receive at least one training point. Similarly, for a height 8 YAHOO tree (best model), only 109 nodes out of total 255 nodes receive at least one training point. These examples indicate that the trees learned by our method can be compressed substantially.

Table 11: Hyperparameter selection for our method: we use default values for many, and cross-validated over the given sets for the last 3.

| HYPERPARAMETER | DEFAULT VALUE or RANGE |
|---|---|
| OPTIMIZER | RMSprop |
| LEARNING RATE | $1e$-2 |
| LEARNING RATE SCHEDULER | CosineLR Scheduler (3 restarts) |
| GRADIENT CLIPPING | $1e$-2 |
| MINI-BATCH SIZE | 128 |
| EPOCHS | 30-400 (depending on dataset size) |
| OVERPARAMETERIZATION ($L$) | $\{1,3\}$ (hidden dimensions in Table 12) |
| REGULARIZATION ($\lambda_1$ OR $\lambda_2$)[16] | $\{5e$-5$, 1e$-5$, 5e$-6$, 1e$-6$, 0\}$ if $L > 1$ else 0 |
| MOMENTUM | $\{0, 0.3\}$ if regression else $\{0, 0.3, 0.8\}$ |

## C.3 Implementation details

### C.3.1 Offline/batch setting

**DGT** We implement the quantization and $\arg\max$ operations in PyTorch as autograd functions. Below, we give details of hyperparameter choices for any given height ($h \in \{2, 4, 6, 8, 10\}$).

We train using RMSProp optimizer with learning-rate $1e$-2 and cosine scheduler with 3 restarts; gradient clipping $1e$-2; momentum of optimizer $\in \{0, 0.3\}$ for regression and $\{0, 0.3, 0.8\}$ for classification; regularization $\lambda_1, \lambda_2 \in \{5e$-5$, 1e$-5$, 5e$-6$, 1e$-6$, 0\}$; and overparameterization $L \in \{1, 3\}$ with hidden space dimensions given in Table 12. We use a mini-batch size of 128 and train our model for 30-400 epochs based on the size of dataset. Table 11 summarizes the choices for all the hyperparameters. Note that we use default values for most of the hyperparameters, and cross-validate only a few.

Table 12: Hidden dimensions for overparameterization ($L = 3$): we use a fixed set of $d_i$'s for a given height $h$; note $d_3 = 2^h - 1$.

| HEIGHT | HIDDEN DIMENSIONS |
|---|---|
| 2 | [240, 240, 3] |
| 4 | [600, 600, 15] |
| 6 | [1008, 1008, 63] |
| 8 | [1530, 1530, 255] |
| 10 | [2046, 2046, 1023] |

**DGT-Forest** We extend DGT to learn forests (DGT-Forest, presented in Section 5.2) using the bagging technique, where we train a fixed number of DGT tree models independently on a bootstrap sample of the training data and aggregate the predictions of individual models (via averaging for regression and voting for classification) to generate a prediction for the forest. The number of tree models trained is set to 30. The sample of training data used to train each tree model is generated by randomly sampling with replacement from the original training set. The sample size for each tree relative to the full training set is chosen from $\{0.7, 0.85, 0.9, 1\}$. Hyperparameters for the individual tree models are chosen as given previously for DGT, with the following exceptions: momentum of optimizer $\in \{0, 0.2, 0.3, 0.4, 0.6\}$ and regularization is used even when $L = 1$.

**TAO** We implemented TAO in Python since the authors have not made the code available yet (Zharmagambetov, April 2021). For optimization over a node, we use scikit-learn's LogisticRegression with the liblinear solver [17]. For the classification datasets, we initialize the tree with CART (trained using scikit-learn) as mentioned in Carreira-Perpinán and Tavallali (2018), train for 40 iterations, where for the LogisticRegression learner, we set maxiter= 20, tol= $5e$-2 and cross-validate $\lambda_1 \in \{1e$-$4, 1e$-$3, 1e$-$2, 1e$-$1, 1, 10, 30\}$. The results we obtain are close to/marginally better than that inferred from the figures in the supplementary material of Carreira-Perpinán and Tavallali (2018).

For the regression datasets we initialize the predicates in the tree randomly as mentioned in Zharmagambetov and Carreira-Perpinan (2020). For PDBBIND, we initialize the leaves randomly in $[0, 1)$, train for 40 iterations, set maxiter $= 20$, tol $= 5e$-2 and cross-validate $\lambda_1 \in \{1e$-$3, 1e$-$2, 1e$-$1, 1\}$. For the large datasets, MICROSOFT and YAHOO, we initialize all leaves to 1, train for 20 iterations with maxiter $= 20$, tol $= 1e$-2 and cross-validate $\lambda_1 \in \{1e$-$3, 1e$-$2, 1e$-$1\}$. In all cases, post-training, as done by the authors, we prune the tree to remove nodes which don't receive any training data points. For the other regression datasets, despite our efforts, we weren't able to reproduce the reported numbers, so we have not provided height-wise results for those (in Figure 4).

**LCN** We use the publicly available implementation[18] of LCN provided by the authors. On all classification and regression datasets we run the algorithm for 100 epochs, using a batch size of 128 (except for the large datasets YEARPRED, MICROSOFT and YAHOO where we use 512) and cross validate on: two optimizers - Adam (AMSGrad variant) and SGD (Nesterov momentum factor 0.9 with learning rate halving every 20 epochs); learning rate $\in \{1e$-$5, 1e$-$4, 3e$-$4, 1e$-$3, 3e$-$3, 1e$-$2, 3e$-$2, 1e$-$1, 3e$-$1, 1\}$; dropout probability $\in \{0, 0.25, 0.5, 0.75\}$; number of hidden layers in $g_\phi \in \{0, 1, 2, 3, 4, 5\}$ and between the model at the end of training vs. the best performing *early-stopped* model. On HIVand PDBBIND we were able to improve on their reported scores.

**TEL** We use the publicly available implementation (TEL). We use the Adam optimizer with mini-batch size of 128 and cross-validate learning rate $\in \{1e$-$5, 1e$-$4, 1e$-$3, 1e$-$2, 1e$-$1\}$, $\lambda_2 \in \{0, 1e$-$8, 1e$-$6, 1e$-$4, 1e$-$3, 1e$-$2, 1e$-$1, 1, 10, 100\}$. The choice of $\gamma \in \{1e$-$4, 1e$-$3, 1e$-$2, 1e$-$1, 1\}$ is explicitly specified in Section 5 when we present the results for this method. To compute the (mean relative) FLOPS, we measure the number of nodes encountered by their model during inference for each test example, and compute the average.

---

[7]https://www.csie.ntu.edu.tw/ cjlin/libsvmtools/datasets/multiclass.html

[8]We use the provided train-test splits; in case a separate validation dataset is not provided we split train in ratio $0.8 : 0.2$

[9]Wu et al. (2018)

[10]https://www.dcc.fc.up.pt/ ltorgo/Regression/DataSets.html

[11]Splits and shuffles obtained from Zharmagambetov and Carreira-Perpinan (2020)

[12]http://www.cs.toronto.edu/ delve/data/comp-activ/desc.html

[13]https://archive.ics.uci.edu/ml/datasets.php

[14]https://www.microsoft.com/en-us/research/project/mslr/

[15]https://webscope.sandbox.yahoo.com/catalog.php?datatype=r

[16]We do not use $L_1$ and $L_2$ regularization simultaneously

[17]The solver doesn't allow learning when all data points belong to the same class. While in many cases this scenario isn't encountered, for MICROSOFT and YAHOO we make the solver learn by augmenting the node-local training set $D$ with $(-\mathbf{x}_i, -y_i)$ for some $i$, where $(\mathbf{x}_i, y_i) \in D$.

[18]https://github.com/guanghelee/iclr20-lcn

**PAT and ablations**  For the probabilistic annealed tree model (presented in Table 1 as well as in Table 7(c)), we use a sigmoid activation on the predicates and softmax activation after the AND layer. We tune the annealing of the softmax activation function with both linear and logarithmic schedules over the ranges [[1,100],[1,1000],[1,10000],[10,100],[10,1000],[10,10000]]. Additionally we tune momentum of optimizer. We follow similar annealing schedules in quantization ablation as well but use additive AND layer and over-parameterization instead.

**CART**  We use the CART implementation that is part of the scikit-learn Python library. We cross-validated min_samples_leaf over values sampled between 2 and 150.

**Ridge**  We use the Ridge implementation that is part of the scikit-learn Python library. We cross-validated the regularization parameter over 16 values between $1e$-6 and $5e$-1.

### C.3.2  Online/bandit setting

In the online/bandit classification setting, we do not perform any dataset dependent hyperparameter tuning for any of the methods. Instead we try to pick hyperparameter configurations that work consistently well across the datasets.

**DGT-Bandit**  For the classification setting we remove gradient clipping and learning rate scheduler. We further restrict to $L = 1$, $\lambda_1 = 0$, $\lambda_2 = 0$, momentum $= 0$ (and mini-batch size $= 1$, by virtue of the online setting and accumulate gradient over four steps). We use the learning rate and $\delta$ parameter which work consistently across all datasets. For the classification case we set $\delta$ (exploration probability in Eqn. (7)) to 0.3 and $\delta$ parameter in Eqn. (10) and Eqn. (9) to 0.5. Learning rate $1e$-3 is used. For the regression setting we only tune the learning rate $\in \{1e$-2,$1e$-3$\}$, $\delta \in \{1,0.5,0.1,0.05,0.01,0.001\}$ and momentum of optimizer.

**Linear Regression**  We implemented linear regression for the bandit feedback setup in PyTorch. We use RMSprop optimizer and we tune the learning rate $\in \{1e$-2$, 1e$-3$\}$, momentum $\in \{0, 0.4, 0.8\}$, regularization $\in \{1, 0.5, 0.1, 0.05, 0.01\}$. We use number of epochs similar to our method. To compute the gradient estimate, we vary $\delta$ (in Eqn. (9)) $\in \{1, 0.5, 0.1, 0.05, 0.01, 0.001\}$.

**$\epsilon$-Greedy (Linear)**  For linear contextual-bandit algorithm with $\epsilon$-greedy exploration (Equation 6 and Algorithm 2 in Bietti et al. (2018)), we use VowpalWabbit (VW) with the --cb_explore and --epsilon flags. We tune learning rate $\eta \in \{0.001, 0.01, 0.1, 1\}$ and the probability of exploration $\epsilon \in \{0.01, 0.1, 0.3, 0.5\}$. We use $\eta = 0.01, \epsilon = 0.5$ which work well across all the datasets.

**CBF**  We use the official implementation provided by the authors (CBF) and use the default hyperparameter configuration for all datasets. Since their method works with only binary features, we convert the categorical features into binary feature vectors and discretize the continuous features into 5 bins, following their work (Féraud et al., 2016).

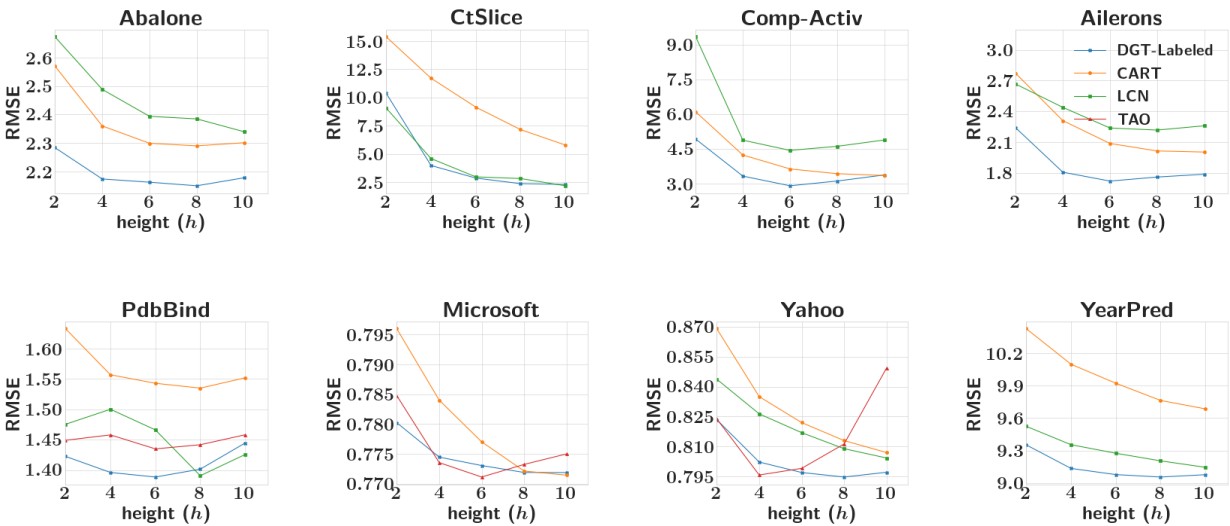

Figure 4: Mean Test RMSE (over 10 random seeds) of competing methods vs tree height for regression datasets. We do not show the performance of TAO method on some datasets (see Sections 5.1 and C.1). LCN did not converge on MICROSOFT (see Section 5.1).

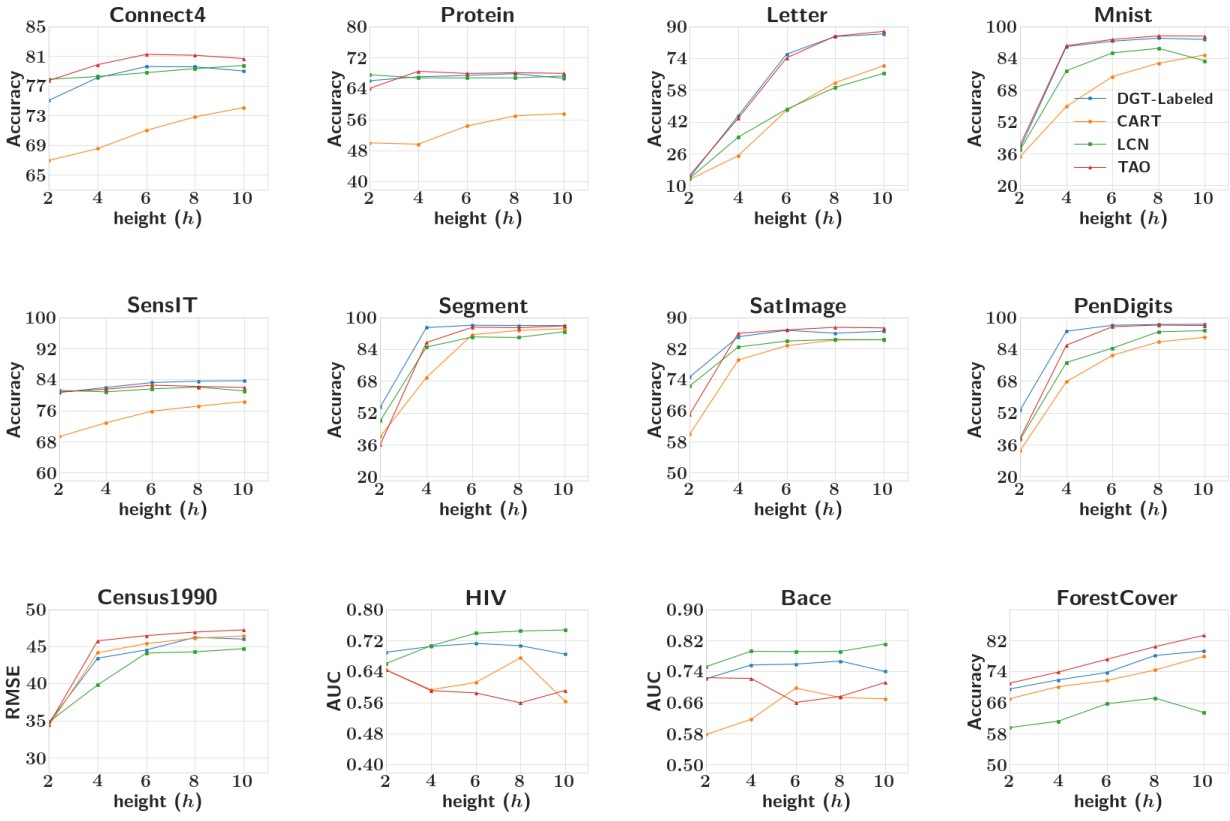

Figure 5: Mean (%) Test Accuracy or Test AUC of competing methods (over 10 random seeds) vs tree height for classification datasets.

