# OpenReview forum: "Learning Accurate Decision Trees with Bandit Feedback via Quantized Gradient Descent"
_TMLR — Accepted by TMLR_

### Review · Reviewer_fNzK · 2022-06-15

**Summary Of Contributions:**

This paper introduces DGT, dense gradient trees, a variation of soft tree learning algorithm. Assuming the initial tree structure is fixed (complete tree) with random parameters at each node, the learning is done via SGD as follows:
- During forward propagation, a tree is considered as hard and only the most probable path is followed per instance;
- As for the backward pass, the tree is considered as soft where gradients of the step function are computed via "straight-through" estimation.

The method is generically applicable to supervised problems such as regression and classification, but the authors also provide extension for Bandit setting where reward/loss function is not explicitly given (but can be computed). Applying SGD for supervised problems is straightforward, whereas in Bandit setting authors use a simple finite difference approximation of derivatives.

**Requested Changes:**

see the previous section. In general, I find that the paper will be interesting and beneficial for some individuals in TMLR's audience and thus, I consider accepting this manuscript if: either claims are reformulated and/or additional experiments/metrics are added based on my comments.

**Strengths And Weaknesses:**

I will evaluate Strengths And Weaknesses based on claims made in the submission and whether those claims are supported by "accurate, convincing and clear evidence" as per reviewer guidelines.

Claims that are supported:
- "enabling sample-efficient tree learning in online, bandit feedback setting without exact information on the loss function or labels." The method is indeed efficient to apply in bandit setting.
- "easily extensible to ensembles that outperform widely-used choices by practitioners." The claim is well-supported experimentally. In terms of methodology, it also makes sense as ensembling trees is relatively straightforward to perform.

Claims that are not supported or weakly supported:
- "Novel, unified, differentiable solution for learning decision trees accurately for practical and standard evaluation settings". I do have some concerns regarding novelty. I think that the method is a variation of soft decision trees with some trivial modifications: "and"->"or"gradients, straight-through gradient estimation which is commonly practiced in quantization literature, etc. I don't mind of re-phrasing this as "novel variation of soft decision trees" or smth similar.
- logarithmic inference cost. Although, this is true for tree part, but over-parameterization adds additional cost which makes the method less "logarithmic" per say compared to conventional CART trees. In order to obtain a strong support for this claim, I'd suggest adding model sizes. This is partially done in fig.1 but only for the inference time and for DGT vs TEL. How about other approaches? CART, TAO? I assume their inference time is faster (especially given the over-parameterization of the proposed method)? I'd suggest adding a proper comparison of model sizes (including inference time) for all models.
- "competitive or superior performance compared to SOTA tree learning methods on a variety of datasets, across different problem settings..." Although I appreciate for including extensive evaluation of the proposed approach, I think that some critical baselines are missing. For instance, since authors use over-parameterization, I think it would be fair to include more advanced baselines which also applies feature transformations (e.g. Kontschieder et al. 2015, Tanno et al. 2019 but with similar model sizes); or TAO with linear leaves? Or maybe report the results with/without over-parameterization?
- "significantly more accurate and sample-efficient than general methods for online settings". I'm not sure what does "general methods for online setting" means exactly, but assuming that soft-trees are considered as such, I don't think that this claim is validated without addressing the previous comment. Moreover, I'd not consider the difference between TEL as "significant" (also why TEL is missing for YAHOO?).

---

> ### Author Response · Authors · 2022-07-04
> **Response to Reviewer fNzK**
>
> > I do have some concerns regarding novelty. I think that the method is a variation of soft decision trees…I don't mind of re-phrasing this as "novel variation of soft decision trees" or smth similar.
>
> As the reviewer rightly points out, our technique is indeed a combination of known simple but effective techniques for learning accurate (hard) decision trees, in a way that naturally applies to the bandit setting. The simplicity and the generalization of our technique to different learning scenarios is what we consider the strength of our work. To the best of our knowledge, we do not know of any state-of-the-art tree learning technique that learns hard (i.e., no probabilistic or soft decision making at inference time) oblique (i.e. uses only a linear model in the internal nodes as in the classical tree formulation, in Definition 1) trees with extensive applicability as ours. We will re-phrase the novelty claim as suggested by the reviewer.
>
> > logarithmic inference cost. Although, this is true for tree part, but over-parameterization adds additional cost which makes the method less "logarithmic" per say compared to conventional CART trees
>
> As discussed in para 3 of Sec 3.2, we use *linear* overparameterization *only* during training. Because all the operations are linear, we obtain a linear model at each tree node, producing the classical oblique (hard) tree exactly as stated in Def 1 at inference (note that this would not be possible had we used non-linear overparameterization during training). Thus, our tree model admits logarithmic inference time --- O(d) computations at each internal node of the tree, along the deterministic root-to-leaf path which is at most O(h) long. In Appendix C.2, we also show that the tree model we learn is typically much sparser than a fully complete binary tree.
>
> > Although I appreciate for including extensive evaluation of the proposed approach, I think that some critical baselines are missing. For instance, since authors use over-parameterization, I think it would be fair to include more advanced baselines which also applies feature transformations (e.g. Kontschieder et al. 2015, Tanno et al. 2019 but with similar model sizes)
>
> As mentioned in the above reply, our comparisons are fair given that we produce standard oblique decision trees of Def 1.
>
> > what “general methods for online setting" means exactly, but assuming that soft-trees are considered as such, I don't think that this claim is validated without addressing the previous comment
>
> Soft-trees are strictly more expressive model class than hard trees (Definition 1), so they are not really a fair comparison. Also, see our response to TEL comparison below in this context.
>
> > Moreover, I'd not consider the difference between TEL as "significant"
>
> Firstly, we present comparisons to TEL only in the supervised setting (not in the bandit setting). Secondly, to make arguments about advantages over TEL, which is a soft tree method, we need to look at the achieved accuracy *for a given* certain inference cost budget. We have enough evidence that suggests that our method is significantly better in this regard. Specifically, based on Fig. 1 (left), we can see that our method is up to 30% more accurate on the *letter* dataset when relative FLOPS is ~1 (i.e., both the models have similar inference complexity and model size). However, for reaching an RMSE equal to what DGT achieves, TEL requires more than 8x inference flops compared to DGT on the *Year* dataset.
>
> > why TEL is missing for YAHOO?
>
> The TEL method does not converge on both Yahoo and HIV datasets within 48 hours and therefore we do not report numbers on those datasets. We will add this note to the paper.

---

> > ### Comment · Reviewer_fNzK · 2022-07-11
> > **Additional comments**
> >
> > Most of my concerns are addressed. Thanks! See additional comments below:
> >
> > > As mentioned in the above reply, our comparisons are fair given that we produce standard oblique decision trees of Def 1.
> > > ...
> > > Soft-trees are strictly more expressive model class than hard trees (Definition 1), so they are not really a fair comparison. Also, see our response to TEL comparison below in this context.
> >
> > I still think that adding those baselines are critical to give the entire picture. Because:
> > - the given method uses soft trees during training and makes it hard once training is done (for inference). It is similar to training soft tree and making it hard afterwards. It is IMPORTANT to have this comparison to compare against "theoretical" maximum.
> > - Adding TAO with linear leaves (at least for some benchmarks) since by the linearity, one can also show that TAO with linear leaves can be also equivalently converted into classical oblique tree.
> > - Alternatively authors can restate their claim: "superior performance compared to SOTA".

---

> > > ### Author Response · Authors · 2022-07-12
> > > **Response to Reviewer fNzK's Additional comments**
> > >
> > > > the given method uses soft trees during training and makes it hard once training is done (for inference). It is similar to training soft tree and making it hard afterwards. It is IMPORTANT to have this comparison to compare against "theoretical" maximum.
> > >
> > > Thank you. We agree that soft trees indeed provide a useful comparison point to our method.  We would like to emphasize that we work with hard predictions (i.e., forward pass) during training, unlike standard soft tree training techniques that work with soft predictions.
> > >
> > > In our experience, training a soft tree first and then making it hard has often yielded poor performance. A good compromise we’ve found is to carry out annealing of the soft decisions as training iterations progress, to gradually bring them close to a hard decision, and as a final step make them completely hard (see 3. in Section 5.4). The comparison of this strategy with the DGT algorithm is presented in column (c) of Table 5, reproduced below.
> > >
> > > | | Annealing        | DGT     |
> > > |---|------|------|
> > > Ailerons |  $1.73 \pm 0.021$ | $1.72 \pm 0.016$ |
> > > Abalone | $2.31 \pm 0.06$ | $2.15 \pm 0.026$ |
> > > YearPred | $9.18 \pm 0.04$ | $9.05 \pm 0.012$ |
> > >
> > > Also note that the state-of-the-art TEL method, with $\gamma = 1$, yields soft trees (i.e., without hardening). As expected, in section 5.2, we find that it outperforms DGT by 2-10% on SatImage, PenDigits, and Letter datasets.
> > >
> > >
> > >
> > > > Adding TAO with linear leaves (at least for some benchmarks) since by the linearity, one can also show that TAO with linear leaves can be also equivalently converted into classical oblique tree.
> > >
> > > Below, we present the RMSE of the two methods - TAO (with linear leaves) and DGT (with constant leaves) on 5 regression datasets. The scores for TAO have been sourced from [1] which provides scores for linear leaves only on regression datasets. However, note that the equivalence of trees with linear leaves and standard oblique trees with constant leaves holds only in the (multi-class) classification setting, not in the regression setting. We will add some results for the classification setting (not reported in [1]) in the revised paper.
> > >
> > > | | TAO (with linear leaves)        | DGT (with constant leaves)    |
> > > |---|------|------|
> > > Ailerons | $1.74 \pm 0.01$ | $1.72 \pm 0.016$
> > > Abalone | $2.07 \pm 0.01$ | $2.15 \pm 0.026$
> > > CompActiv | $2.58 \pm 0.02$ | $2.91 \pm 0.149$
> > > CtSlice | $1.16 \pm 0.02$ | $2.30 \pm 0.166$
> > > YearPred | $9.08 \pm 0.03$ | $9.05 \pm 0.012$
> > >
> > > [1] Zharmagambetov A. ; Carreira-Perpinan M. A.. Smaller, More Accurate Regression Forests Using Tree Alternating Optimization. (ICML 2020)

---

> > > > ### Comment · Reviewer_fNzK · 2022-07-12
> > > > **Good to go**
> > > >
> > > > Thanks for additional comparison tables! Please add the last table to the main paper (comparison with TAO linear leaf).
> > > >
> > > > > Also note that the state-of-the-art TEL method, with $\gamma=1$, yields soft trees (i.e., without hardening). As expected, in section 5.2, we find that it outperforms DGT by 2-10% on SatImage, PenDigits, and Letter datasets.
> > > >
> > > > Please add this as an additional table and/or figure. And please restate your claim regarding SOTA performance. Otherwise, I'm satisfied with authors' responses and have no further comments. Congratulations on your fine work!

---

### Review · Reviewer_tDsy · 2022-06-16

**Summary Of Contributions:**

The authors consider the problem of learning hard decision trees (where each input example lands in a single leaf) with bandit feedback. Key contributions include:

- They re-formulate the tree learning problem such that 1) the multiplicative form is rewritten as sums so as to control ill-conditioned gradients; 2) the additions of deep linear layers are allowed.

- They propose a new algorithm to solve the proposed formulation. The algorithm uses quantized gradient descent.

- They conduct experiments in both supervised learning settings and bandit settings. In the supervised setting, the empirical performance is roughly comparable to TAO. In the bandit setting, the proposed algorithm outputs a more accurate model faster than basic benchmarks.

**Broader Impact Concerns:**

No.

**Requested Changes:**

- (Particularly Critical) Please address the concerns regarding the bandit experiments raised in "weaknesses".

- (Critical) To handle bandit feedback in the classification setting, the gradient (8) is computed based on the arm-sampling scheme specified in (7). Does that mean we have to use $\epsilon$-greedy type of algorithms? Can we use more sophisticated bandit algorithms there? Please clarify.

- (Critical) In Section 3.2, the authors only discuss linear layers. However, based on my knowledge, many often-used layers are not linear, and I don't think the current method can be easily extended. Can authors discuss how limited this assumption is? I'm fine if extensions can't be made, but I would like the authors to at least acknowledge it.

- Please define "FLOPS" in Section 5.2.


**Strengths And Weaknesses:**

**Strengths**

- The paper is overall well-structured and easy-to-follow.

- The arguments in the main paper overall make sense to me.

**Weaknesses**

- I think the experiments for the bandit feedback setting is a bit confusing. Some of my questions are listed below.
  - The authors evaluate the different methods based on "the accuracy/RMSE computed on a fixed held-out set for the model obtained after n rounds". This is very different from the average regret defined in Section 2, and in essence closer to "best-arm identification". At least for classification problems, is it possible to include bandit algorithms designed for best-arm identification? That looks like a more reasonable comparison.
  - The benchmark policy is $\epsilon$-greedy with linear policies. Is it possible to compare $\epsilon$-greedy with more complicated policy classes? The inferior performances of $\epsilon$-greedy with linear may be only because the linear class is too small to characterize the true model; besides, these $\epsilon$-greedy curves converge much faster.
  - Please specify the $\epsilon$ used in the experiments for $\epsilon$-greedy. Also, what's the impact of choosing different $\epsilon$'s?

---

> ### Author Response · Authors · 2022-07-04
> **Response to Reviewer tDsy**
>
> > The authors evaluate the different methods based on "the accuracy/RMSE computed on a fixed held-out set for the model obtained after n rounds". This is very different from the average regret defined in Section 2, and in essence closer to "best-arm identification". At least for classification problems, is it possible to include bandit algorithms designed for best-arm identification? That looks like a more reasonable comparison.
>
> We find that the convergence behavior on training data (which corresponds to average regret defined in Sec 2) is similar to the convergence behavior on the held-out (test) data. In practice, we would want the expected loss on *unseen (held-out)* examples to be small for ML models in deployment.
>
> Additionally (and importantly), we are not just interested in the “final loss” but rather the loss at each time step (which is crucial in deployment, and is what we focus on in our paper). From this perspective, it seems incorrect to pose this as a “best-arm identification” problem.
>
> > The benchmark policy is $\epsilon$-greedy with linear policies. Is it possible to compare $\epsilon$-greedy with more complicated policy classes?
>
> Our goal is to provide a widely-used online method (that is part of the popular VW library) as a baseline. We could use complex neural RL policies that may be able to model the problem better at convergence, but it would not make a meaningful or fair comparison when our goal is to make predictions *efficiently* (via a tree model) at each time step.
>
> > Please specify the $\epsilon$ used in the experiments for $\epsilon$-greedy
>
> For all the relevant methods, we choose a fixed $\epsilon$ (from a range of values) that achieves good validation performance on two datasets. We discuss this in Appendix C.3.2 under the respective methods.
>
> > To handle bandit feedback in the classification setting, the gradient (8) is computed based on the arm-sampling scheme specified in (7). Does that mean we have to use $\epsilon$-greedy type of algorithms? Can we use more sophisticated bandit algorithms there? Please clarify
>
> Yes, in our method, the $\delta$ in Eqn (7) plays the role of $\epsilon$. However, we can also use "UCB style of ideas" where we determine how to the trade-off exploration vs. exploitation using confidence bounds (see Zhou et al., 2020) or "Thompson-sampling style of ideas" where we compute the posterior distribution of each arm being optimal to sample the arms (see Zhang et al., 2021)
>
> > In section 3.2, the authors only discuss linear layers. However, based on my knowledge, many often-used layers are not linear, and I don’t think the current method can be easily extended. Can authors discuss how limited this assumption is? I’m fine if extensions can’t be made, but I would like the authors to at least acknowledge it.
>
> The reason we stick to linear overparameterization is that it enables us to learn the standard oblique trees of Definition 1 (as clarified in para 3 of Sec 3.2). However, we can allow non-linear overparameterization which will lead to models much more complex than standard trees, and our technique does extend to this setting in a straightforward manner. The BackProp Alg 2 used in Line 10 of Alg 1 can be implemented for any non-linear parameterization (we only rely on the fact that the model class used for computing **a** in Line 7 of Alg 2 is differentiable). We have alluded to this in the last para of Sec 2. But, we have not empirically tried more complex parameterization.
>
> > Please define FLOPS in Section 5.2
>
> FLOPS is floating-point operations. We use a simple scheme to count FLOPS for all the compared methods. The total number of FLOPS to compute the prediction on a given test example of d features is O($d$) times the number of nodes in the tree visited during inference (which is h for our method, O($2^h$) for soft trees, and often larger than h for TEL).
>
> Zhou, D. ; Li, L. ; Gu, Q. Neural Contextual Bandits with UCB-based Exploration. (ICML 2020).
>
> Zhang, W. ; Zhou, D. ; Li, L. ; Gu, Q. Neural Thompson Sampling. (ICLR 2021).

---

### Review · Reviewer_HdAq · 2022-06-21

**Summary Of Contributions:**

This paper proposes a learning method for decision trees without softening the trees. This paper considers both the traditional supervised learning and bandit-feedback setting.

**Requested Changes:**

Add a background section to clarify
(1) the motivation about why the bandit setting is focused
(2) related concepts about the bandit.

**Strengths And Weaknesses:**

Strengths.
The proposed method looks interesting and reasonable. Experiments are extensive with many datasets and baselines.


Weaknesses.
(1) I feel like the paper can be decomposed into two separate parts: the first one is about using the straight-through estimator to handle the “hard” step functions, the second one is how to estimate the gradient for the bandit-feedback setting.
(2) It would be better if the authors could elaborate on why they want to focus on the bandit-feedback setting rather than the traditional supervised setting. I think the motivation should be made more clear.
(3) The techniques (e.g., straight-through and gradient estimation) used in this paper are simply taken from existing papers. The novelty of this paper is limited in this sense.
(4) The structure of the paper can be improved. For example, Algorithm 2 seems to be integral to understanding Algorithm 1. But Algorithm 2 is put to appendix, making the main paper not self-contained.
(5) Background about bandit should be more complete. The current terminology regarding bandit is confusing: contextual bandit setting, bandit feedback, online optimization, black-box loss function. These concepts are related but certainly not the same. For example, contextual bandit is defined as the multi-armed bandit setting where we have a contextual vector available at each round, whereas bandit feedback is when we can only get the partial feedback regarding the loss of the current state. The bandit feedback is also applied to non-contextual bandit setting. I think it’s better if the authors could have a background section to focus on bandit related concepts.

---

> ### Author Response · Authors · 2022-07-04
> **Response to Reviewer HdAq**
>
> > It would be better if the authors could elaborate on why they want to focus on the bandit-feedback setting rather than the traditional supervised setting. I think the motivation should be made more clear.
>
> The bandit learning setting is a long-studied problem both in the theoretical/optimization communities as well as in the ML community (e.g, [Auer et al., 2002; Dani et al., 2008; Dudik et al., 2011; Shamir, 2013; Agarwal et al., 2014; Bietti et al., 2021]). It is often found in real-world scenarios making it equally relevant from a practical standpoint. For instance, the Vowpal Wabbit (VW) library is extensively used by practitioners for solving learning problems in the bandit setting (e.g., recommender systems with click-feedback). Decision trees, a powerful and efficient class of algorithms, have primarily been studied in the traditional supervised setting but largely ignored in the bandit scenario. Therefore we choose to focus on the latter. We will make this explicit in the Intro.
>
> > The techniques (e.g., straight-through and gradient estimation) used in this paper are simply taken from existing papers. The novelty of this paper is limited in this sense.
>
> We agree with the reviewer. However, as we responded to Reviewer fNzK, our technique is indeed a combination of known simple but effective techniques for learning accurate (hard) decision trees, in a way that naturally applies to the bandit setting. The simplicity and the generalization of our technique to different learning scenarios is what we consider the strength of our work. To the best of our knowledge, we do not know of any state-of-the-art tree learning technique that learns hard (i.e., no probabilistic or soft decision making at inference time) oblique (i.e. uses only a linear model in the internal nodes as in the classical tree formulation, in Definition 1) trees with extensive applicability as ours.
>
> > The structure of the paper can be improved. For example, Algorithm 2 seems to be integral to understanding Algorithm 1. But Algorithm 2 is put to appendix, making the main paper not self-contained.
>
> We view this differently. Algorithm 1 is indeed mostly self-contained and helps the reader appreciate the core idea of using hard trees in the forward pass, but using dense gradients in the backward pass without getting lost in the gradient computation algebra. Algorithm 2 is mostly technical detail which is why we deferred to the Appendix --- standard back-propagation procedure with some vector calculus, and application of the straight-through estimator trick as described in Section 4a. We will try to fit Algorithm 2 in the main paper.
>
> > Background about bandit should be more complete…
>
> We agree with the reviewer that terminology can be confusing, although the related literature does use some of these terms interchangeably --- we will make the distinctions clear or stick to a single terminology throughout.

---

> > ### Author Response · Authors · 2022-08-10
> > **Hope we clarified your concerns**
> >
> > Looking forward to hearing from the reviewer on our responses. Please let us know if you've any follow-up questions.

---

### Author Response · Authors · 2022-07-19
**Summary of Changes in the Revision**

Based on the feedback given by all reviewers we have updated the paper, with changes highlighted in blue.

Summary of changes:

**Reviewer HdAq**
- Motivation for focusing on bandit setting has been made more explicit in Page 1.
- Algorithm 2 has been moved to the main paper (Page 7).
- Terminology regarding bandits and the setting we focus on, has been made clearer (Page 1 and 2). Throughout the paper, we've ensured to sticking with a consistent terminology.

**Reviewer tDsy**
- Added a remark in Page 5 which clarifies that despite over-parameterization DGT learns standard oblique trees with linear models in internal nodes.
- Defined FLOPS in Page 10.

**Reviewer fNzK**
- Re-phrased novelty claim in Abstract and Page 2.
- Added Table 3 (Page 10) that compares DGT with TEL in a tabular form.
- Added a section (and table) on comparison with trees with linear leaves (Page 11).
- Added note for why TEL is missing for Yahoo (Page 9).

---

### Author Response · Authors · 2022-09-08
**Uploaded Camera Ready Version**

We thank the AE for all their suggestions. A camera ready version of the paper has been uploaded after incorporating these.

We would once again like to thank all reviewers for taking the time to review our paper.

---

### Decision · Action_Editors · 2022-08-12

**Recommendation:** Accept with minor revision

**Comment:**

This is definitely an interesting work which, while drawing from several known ideas, successively synthesizes them into the new, widely applicable method that uses hard trees in order to achieve state-of-the-art performance in general and performance superior to SOTA in the bandit setting. Extensive empirical evaluations comparing against many state-of-the-art methods attests to the competitiveness/superiority of DGT. All the reviewers support the acceptance of the paper. This work is a welcome contribution to TMLR. Congratulations!

I still would like to suggest some very minor revisions (a few of them involve typos):
- p1, second paragraph: "Bandit setting" —> "The bandit setting"
- p2, first line "supervised setting" —> "the supervised setting"
throughout the paper: it is better to write out "wrt" as "with respect to"
- p2, last paragraph before Related Work: "in online, bandit feedback setting" —> "in the online, bandit feedback setting"
- I believe your "Related Work" header is written as \\paragraph{}, and this doesn't work because (to my eyes) the subsequent headers "Non-greedy techniques:", etc. are also written as \\paragraph{} (meaning that they are not interpreted as belonging to the "Related Work" section. You should be able to come up with a fix for this.
- p2, paragraph before "Non-greedy techniques:": "Most relevant ones" —> "The most relevant ones"
- p2 and p3: "backprop" —> "backpropagation" (to be consistent with other instances where you did write "backpropagation"
- p3: "mixed ILP" —> "mixed integer linear program (ILP)"
- p3: "The recently proposed Zantedeschi et al. (2021)" —> "The recently proposed (Zantedeschi et al., 2021)"
- p3: "that yields \\emph{small} number" —> "that yields a \\emph{small} number"
- p3, "Bandit feedback:": "learns axis-aligned decision tree/forest" —> "learns an axis-aligned decision tree/forest"
- p3: Just before the math display prior to (1) (and again in the sentence before (1)), when you define $\\ell$ using $\\mapsto$, I do not think this is the correct use of $\\mapsto$. The RHS of $\\mapsto$ is usually a value, not the range (output space) of the function. I cannot suggest an easy fix but you might see one.
- p3, the second before equation (1): You should be careful in writing $\\{x_i, y_i\\}_{i=1}^n$ as the training data is not a set (the same example can appear multiple times).

Using $(x_i, y_i)_{i=1}^n$
would be better as the training "set" is in fact a sequence.
- be consistent on "overparameterization" vs "over-parameterization"
- There is a typo in the line immediately below Algorithm 1 where you define the sigmoid function: "$\\exp(-s . a)$" perhaps should be "$\\exp(-s \\cdot a)$" ?
- At the end of equation (8), second line, you have a double comma ", ,"
- I noticed that you refer to $L$ as "overparameterization". Perhaps you can clarify this use of the term to refer to $L$ when you first introduce $L$, as it is not that intuitive (it feels more natural to refer $L$ as the number of [hidden] layers)
- In footnote 2, it is not clear what is meant by "cut-off period"; how long was this period?
- In Figure 1 (left), there should be a way to fix the figure/legend so that the legend doesn't invade the curves being plotted!
- In the definition of Huber loss, I believe you are missing a factor of 1/2 in the first case (should be $\\frac{1}{2} (\\hat{y} - y)^2$ if $|\\hat{y} - y| \\leq \\xi$)
- Of course this is up to you, but it would look better if your figures were vector graphics (save as eps or pdf using matplotlib or ggplot etc.)